# Elucidating the tunability of binding behavior for the MERS-CoV macro domain with NAD metabolites

Meng-Hsuan Lin[1], Chao-Cheng Cho[1,2], Yi-Chih Chiu[1], Chia-Yu Chien[2,3], Yi-Ping Huang[4], Chi-Fon Chang[4] & Chun-Hua Hsu [1,2,3✉]

The macro domain is an ADP-ribose (ADPR) binding module, which is considered to act as a sensor to recognize nicotinamide adenine dinucleotide (NAD) metabolites, including poly ADPR (PAR) and other small molecules. The recognition of macro domains with various ligands is important for a variety of biological functions involved in NAD metabolism, including DNA repair, chromatin remodeling, maintenance of genomic stability, and response to viral infection. Nevertheless, how the macro domain binds to moieties with such structural obstacles using a simple cleft remains a puzzle. We systematically investigated the Middle East respiratory syndrome-coronavirus (MERS-CoV) macro domain for its ligand selectivity and binding properties by structural and biophysical approaches. Of interest, NAD, which is considered not to interact with macro domains, was co-crystallized with the MERS-CoV macro domain. Further studies at physiological temperature revealed that NAD has similar binding ability with ADPR because of the accommodation of the thermal-tunable binding pocket. This study provides the biochemical and structural bases of the detailed ligand-binding mode of the MERS-CoV macro domain. In addition, our observation of enhanced binding affinity of the MERS-CoV macro domain to NAD at physiological temperature highlights the need for further study to reveal the biological functions.

[1] Genome and Systems Biology Degree Program, National Taiwan University and Academia Sinica, Taipei 10617, Taiwan. [2] Department of Agricultural Chemistry, National Taiwan University, Taipei 10617, Taiwan. [3] Institute of Biochemical Sciences, National Taiwan University, Taipei 10617, Taiwan. [4] Genomics Research Center, Academia Sinica, Taipei 11529, Taiwan. ✉email: andyhsu@ntu.edu.tw

Nicotinamide adenine dinucleotide (NAD), a dinucleotide consisting of adenine-ribose and nicotinamide-ribose joined by a pair of bridging phosphate groups, is a coenzyme found in all living cells[1]. The cellular pool of free NAD is balanced between synthesis and degradation. The metabolic product, nicotinic acid mononucleotide, can be recycled for NAD synthesis. Adenine nucleotides, including adenosine triphosphate (ATP), adenosine diphosphate (ADP), and adenosine monophosphate (AMP), are the side products of the NAD metabolism pathway[2]. NAD-consuming enzymes such as poly-ADP-ribosyl polymerases, also called Diphtheria toxin-like ADP-ribosyl-transferases (ARTDs), use NAD as a group donor. ARTDs produce PAR by polymerizing the ADPR moieties and releasing the nicotinamide groups from NAD. PAR, as one of NAD metabolites, constructs the backbone of ADP-ribosylation, a kind of reversible chemical modification into protein[3], DNA[4], and RNA[5].

ADP-ribosylation has been suspected of having a dual pro-antiviral role in regulating the virulence of diverse viral families. Several viruses requiring ARTD activity for efficient replication have been discovered by using an ARTD1 inhibitor such as vaccinia virus, herpes simplex virus, JC virus, and porcine reproductive and respiratory syndrome virus[6–9]. However, activation of ADP-ribosylation by viral infection often resulted in apoptosis of the host cell, considered a host immune defense[10–12]. The first discovered antiviral ADP-ribosylation enzyme was ARTD13, which was found to bind viral RNA, inhibit viral replication, induce transcription of viral defensing interferons, activate proteasome degradation, and promote apoptosis[10,13–16]. Also, the antiviral effects of mono-ADP-ribosylation enzymes have been reported[17]. By transcriptome and protein interactome analyses, several mono-ADP-ribosyltransferase, diphtheria toxin-like enzymes (ARTD 10, 12, 14) were also found to inhibit viral replication[18,19]. Positive feedback loops of the expression of these three ARTDs are activated by interferons (IFNs) during infection with the Venezuelan equine encephalitis virus (VEEV)[18,20]. Human sirtuins are NAD-dependent deacylases/mono-ADP ribosyltransferases. By siRNA knockdown and sirtuins inhibitors, all sirtuins were found to have broad-range antiviral properties for both DNA and RNA viruses[21]. In addition, ARTD11 was found upregulated after virus infection and thus promoted IFN-I antiviral response by mono-ADP-ribosylating ubiquitin E3 ligase[22]. So viruses might have developed mechanisms to regulate the ADP-ribosylation-related conflict between viral replication and host immunity.

The ADPR moieties in PAR are recognized by some protein modules, including macro domains[23,24]. The macro domain is highly evolutionarily conserved and has been found widely distributed throughout all kingdoms of life, including in some positive-strand RNA viruses (Alphavirus, Coronavirus, Hepatitis E virus, and rubella virus) (Supplementary Fig. S1). Attention to the function of the macro domain in viral replication has been growing. The ADPR binding and ADPR hydrolase activities of the macro domain from various viruses have been investigated with viral replication[5,25,26]. In addition, increasing interest has been given to NAD metabolism during immune responses, thereby indicating that its modulation is relevant to host–pathogen interactions[27,28]. To what extent viral macro domains interfere in the host immunity by fluctuating the balance of the pool of NAD metabolites by interactions with its ligands still needs to be investigated[10,17,24,29].

As a human pathogen first identified in Saudi Arabia in 2012[30], the Middle East respiratory syndrome-coronavirus (MERS-CoV) harbors a macro domain in its non-structural protein 3 inside the open reading frame 1a (ORF1a)[31,32]. Proteins in ORF1a are related to viral replication of MERS-CoV, and polymorphism in ORF1a is considered to influence cross-host transmission and

evolution of this virus[31]. The MERS-CoV macro domain is considered to possess the enzyme activity of ADPR-1"-monophosphatase[33,34], similar to several viral macro domains[35,36]. Some studies indicated that the viral macro domains could also catalyze ADPR modifications, including severe acute respiratory syndrome coronavirus (SARS-CoV), human coronavirus 229E (HCoV229E), chikungunya virus (CHIKV), and VEEV[25,37–39]. Mutagenesis analyses of several viral macro domains suggested the role of viral defense toward host cellular immunity because the mutated macro domain reduced viral titers in cultured host cells[39–42]. In addition, the N1040A mutation of the nsp3 macro domain from SARS-CoV and HCoV229E showed increased sensitivity of antiviral IFN from host cells[43]. However, the critical piece of the puzzle for the molecular function of viral macro domains is still missing.

We have knowledge of macro domains interacting with NAD-derived metabolites, such as ADPR and PAR, from various species[23,44,45]. The crystal structure of the MERS-CoV macro domain binding with ADPR was solved and showed an α/β globular structure with the hydrophobic cleft able to accommodate a ligand[46]. In the ligand binding site of MERS-CoV macro domain, conserved D20 made a critical recognition to the nitrogen of the adenine group of ADPR. G46, I47, S126, G128, I129, and F130 of MERS-CoV macro domain formed hydrogen bonds with the diphosphate groups of ADPR. Furthermore, A37, N38, K42, G44, A48, V93, and G95 formed hydrogen bonds with the distal ribose of ADPR. The snapshot of the ADPR-complex structure threw light on the interaction between the MERS-CoV macro domain and single free ADPR. However, how the MERS-CoV macro domain interacts with NAD metabolites and ADPR moieties buried inside PAR is still unclear. In addition, some viral macro domains possess RNA-binding ability[25,47]. The existence of NAD-capping RNA has been discovered in prokaryotes, yeast, and eukaryotes[48–51]. Although canonical capping modification of viral RNA involves methylation for efficient replication[52,53], NAD-capping RNA may have potential in the virus shell.

Here, we used a variety of structural tools and additional biophysical experiments to systematically probe the possible ligand binding mechanism of the MERS-CoV macro domain with NAD metabolites. The MERS-CoV macro domain demonstrated a ligand-binding mechanism by regulating the flexible loop motifs surrounding the ligand-binding site (loop β3-α2 and β6-α5). Furthermore, the binding behavior of the MERS-CoV macro domain was altered at physiological temperature especially when binding to NAD metabolites with larger chemical structure than ADPR, which suggests possible thermal activation for viral infection.

## Results

**Evaluation of MERS-CoV macro domain binding with various NAD metabolites**. To elucidate the impact of individual chemical moieties of ligand, differential scanning fluorometry (DSF), a fluorescence-based thermal shift assay, was first used to monitor the binding ability of the protein with various NAD metabolites, including NAD, ADPR, ATP, ADP, and AMP (Fig. 1A). The MERS-CoV macro domain showed enhanced thermal stability, as evidenced by the shift of melting temperature (Tm) from 41.5 °C for the apo form to 49.0, 46.5, 43.0, 44.5, and 42.5 °C, on binding with 100-fold concentrations of ADPR, NAD, ATP, ADP, and AMP, respectively (Fig. 1B). The thermal shifts with the increasing concentrations of NAD metabolites were also tested (Supplementary Fig. S2). In addition, the thermal melting profiles of the MERS-CoV macro domain monitored at 220 nm using circular dichroism (CD) were greatly affected by the presence of these NAD metabolites. The melting temperature of the MERS-

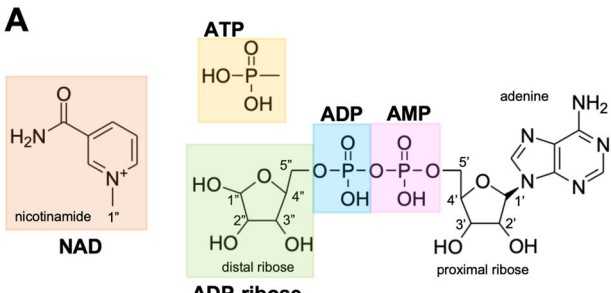

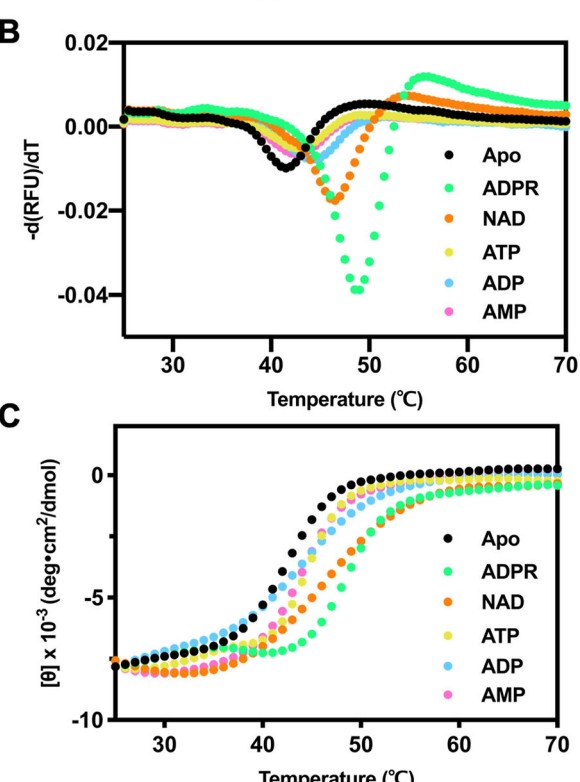

**Fig. 1 Evaluation of MERS-CoV macro domain binding with NAD metabolites by Tm values. A** Chemical structures of NAD metabolites tested in this study. Melting temperatures of 10 μM MERS-CoV macro domain in complex with 1 mM NAD metabolites were determined by **B** thermal shift assays and **C** circular dichroism. MERS-CoV macro domain in the apo form, ADPR-bound, NAD-bound, ATP-bound, ADP-bound, and AMP-bound form are labeled in black circles, green squares, orange triangles, yellow triangles, blue diamonds, and pink circles, respectively.

**Table 1 Melting temperature of the MERS-CoV macro domain in apo form and in complex with various NAD metabolites determined by differential scanning fluorometry (DSF) and circular dichroism (CD).**

|  | Apo | ADPR | NAD | ATP | ADP | AMP |
|---|---|---|---|---|---|---|
| DSF | 41.5 | 49.0 | 46.5 | 43.0 | 44.5 | 42.5 |
| CD | 42.0 | 48.8 | 46.6 | 44.4 | 43.7 | 44.0 |

Unit: °C

CoV macro domain (Tm ~42 °C) was shifted around 1–7 °C (Fig. 1C). The increase in Tm of the MERS-CoV macro domain incubated with various NAD metabolites determined by the two approaches indicated ligand binding ability in the order of ADPR > NAD > ADP > ATP ≅ AMP (Table 1).

**Coordination of NAD metabolites in MERS-CoV macro domain**. The crystal structure of the MERS-CoV macro domain in complex with ADPR was solved (PDB ID 5DUS)[46], presenting an α/β globular structure with a hydrophobic cleft for ligand binding (Supplementary Fig. S3A), similar to other solved macro domains from diverse species. Of note, some macro domains were investigated for interactions with ADPR, ATP, ADP, or AMP but not NAD[29,45,54,55]. To clarify the role of certain residues in ligand recognition, protein crystals of the MERS-CoV

macro domain were soaked with NAD, ATP, ADP, and AMP for detailed observation (Supplementary Fig. S3) and refined to a resolution of 1.68 Å, 2.47 Å, 1.50 Å, and 1.71 Å, respectively (Table 2). The crystal structures of protein complexes were almost identical, but two main structural divergences appeared between two loops (distinguished loop 1, β3-α2 (43HGGGI47) and distinguished loop 2, β6-α5 (126SAGIF130)) of the ligand-binding sites (Fig. 2A). As expected, these two loops presented a closer conformation of binding pocket than other metabolites upon ADPR binding. The binding pocket upon NAD or ATP binding was more like an open state due to an extra nicotinamide or a phosphate group replacement of the distal ribose. However, the ADP-bound macro domain presented a closed state like the ADPR-bound form, and the AMP-bound form was an open state (Fig. 2B). Notably, we first determined the MERS-CoV macro domain in complex with NAD; NAD was previously considered not to interact with macro domains.

To unveil details about the selective modes of ligand recognition, we analyzed the crystal structures of a series of NAD metabolites bound to the MERS-CoV macro domain (Supplementary Fig. S4). The ligand-binding pocket could be classified as three regions: region 1, near the adenine group of the ligand; region 2, constituted by distinguished loops 1 and 2 having a critical role in ligand binding; and region 3, near the distal ribose of the ligand (Supplementary Fig. S5). The similarity between the coordination of ADPR and NAD inside the MERS-CoV macro domain binding pocket extended to the pyrophosphate group (Fig. 3A, B). In the ADPR complex, residues at region3, A36, N38, and H43, formed hydrophobic interactions with the distal ribose of ADPR. Therefore, the side-chain amide of N38 at region3 in the NAD complex connected to the 3″-hydroxyl group of NAD instead. Furthermore, the distal ribose of ADPR configured a T-shaped stacking with F130. Moreover, the distal ribose of NAD was slightly anchored, and the nicotinamide pointed out of the binding pocket (Fig. 3B and Supplementary Fig. S3B). Because of the lack of distal ribose, ATP, ADP, and AMP did not connect with residues, including N38, K42, H43, G44, and G45 (Fig. 3C–E).

Hydrogen bonds that linked phosphate groups from ADPR or ADP with G128 and I129 resulted in a rotation of the side chain of I129, thereby introducing a closed state of ligand binding pocket (Fig. 2B). To describe the open and closed conformations of the binding pocket, we measured the distance between distinguished loops 1 and 2 (distance between the backbone amide nitrogen of G46 and Cδ of I129). The distances for the ADPR-complex and ADP-complex forms as a closed state were 3.58 Å and 4.13 Å, respectively. However, NAD-complex, ATP-complex, and AMP-complex forms presented open pockets with a distance of 7.40 Å, 6.53 Å, and 7.43 Å, respectively (Supplementary Fig. S4). Although these three molecules all make a macro domain present an open state, the AMP-complex had a distinct loop2 conformation, and more water molecules occupied on the binding pocket from ATP and NAD bound forms.

**Table 2 Data collection and refinement statistics.**

|  | NAD-bound MERS-CoV macro domain | ATP-bound MERS-CoV macro domain | ADP-bound MERS-CoV macro domain | AMP-bound MERS-CoV macro domain |
|---|---|---|---|---|
| *Data collection* |  |  |  |  |
| Space group | P12$_1$1 | P12$_1$1 | P12$_1$1 | P12$_1$1 |
| *Cell dimensions* |  |  |  |  |
| $a, b, c$ (Å) | 41.4, 39.3, 45.1 | 41.0, 39.1, 45.1 | 41.4, 39.4, 45.5 | 41.4, 39.5, 45.5 |
| $\alpha, \beta, \gamma$ (°) | 90, 102.1, 90 | 90, 101.6, 90 | 90, 102.3, 90 | 90, 102.4, 90 |
| Resolution (Å)[a] | 28.21–1.68 (1.74–1.68) | 28.03–2.47 (2.56–2.47) | 28.24–1.50 (1.55–1.50) | 29.54–1.71 (1.77–1.71) |
| $R_{merge}$(%) | 8.228 (49.70) | 7.208 (27.95) | 6.71 (43.67) | 6.51 (41.46) |
| $I / \sigma I$ | 14.34 (4.49) | 20.48 (3.68) | 14.41 (2.46) | 14.94 (2.01) |
| Completeness (%) | 97.11 (87.10) | 88.93 (73.59) | 99.23 (93.92) | 95.50 (83.64) |
| Redundancy | 3.7 (3.6) | 3.1 (2.3) | 4.3 (4.3) | 2.6 (2.4) |
| CC1/2 | 0.992 (0.841) | 0.995 (0.884) | 0.998 (0.844) | 0.996 (0.743) |
| CC* | 0.998 (0.956) | 0.999 (0.969) | 1.000 (0.957) | 0.999 (0.923) |
| *Refinement* |  |  |  |  |
| Resolution (Å) | 28.21–1.68 | 28.03–2.47 | 28.24–1.50 | 29.54–1.71 |
| No. reflections | 57105 | 14527 | 100356 | 36362 |
| $R_{work}$ / $R_{free}$ | 0.158/0.195 | 0.170/0.235 | 0.151/0.177 | 0.1570.191 |
| *No. atoms* |  |  |  |  |
| Protein | 1219 | 1237 | 1228 | 1264 |
| Ligand/ion | 44 | 31 | 27 | 23 |
| Water | 230 | 23 | 191 | 198 |
| *B-factors* |  |  |  |  |
| Protein | 15.30 | 30.00 | 16.30 | 15.80 |
| Ligand/ion | 21.50 | 46.40 | 14.40 | 16.20 |
| Water | 30.30 | 30.90 | 31.40 | 29.70 |
| *R.m.s. deviations* |  |  |  |  |
| Bond lengths (Å) | 0.006 | 0.015 | 0.013 | 0.009 |
| Bond angles (°) | 0.92 | 1.59 | 1.50 | 1.33 |

[a]Values in parentheses are for highest-resolution shell.

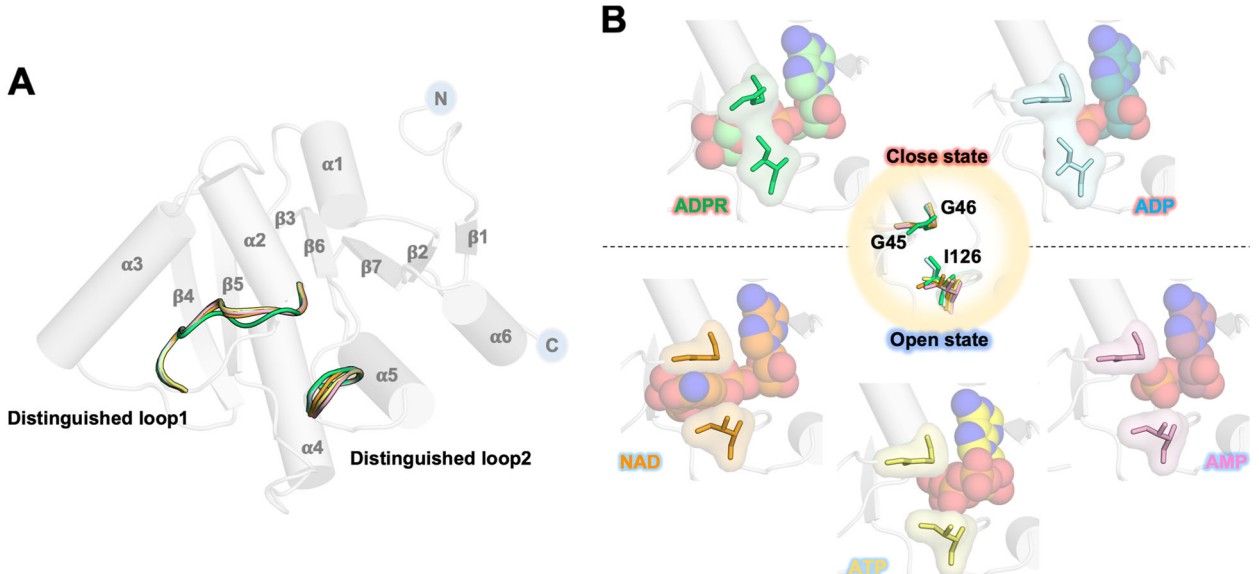

**Fig. 2 Two divergences between the five structures of the MERS-CoV macro domain in complex with different NAD metabolites. A** Two distinguished loops in protein structures complexed with ADPR, NAD, ATP, ADP, and AMP are represented by ribbon models colored in light green, orange, yellow, cyan, and light pink, respectively. **B** The closed/open states of the ligand-binding pocket result from the orientation of residues, G45, G46, and I126, located on the two distinguished loops.

**Identification of critical residues and binding ability of MERS-CoV macro domain by nuclear magnetic resonance (NMR) and isothermal titration calorimetry (ITC) titrations.** To explore the ability of the MERS-CoV macro domain to bind NAD metabolites and elucidate the impact of individual chemical moieties on the interaction, NMR chemical shift perturbation (CSP) experiments were monitored by the acquisition of $^1$H, $^{15}$N HSQC spectra at 298 K. NMR resonance assignment of the

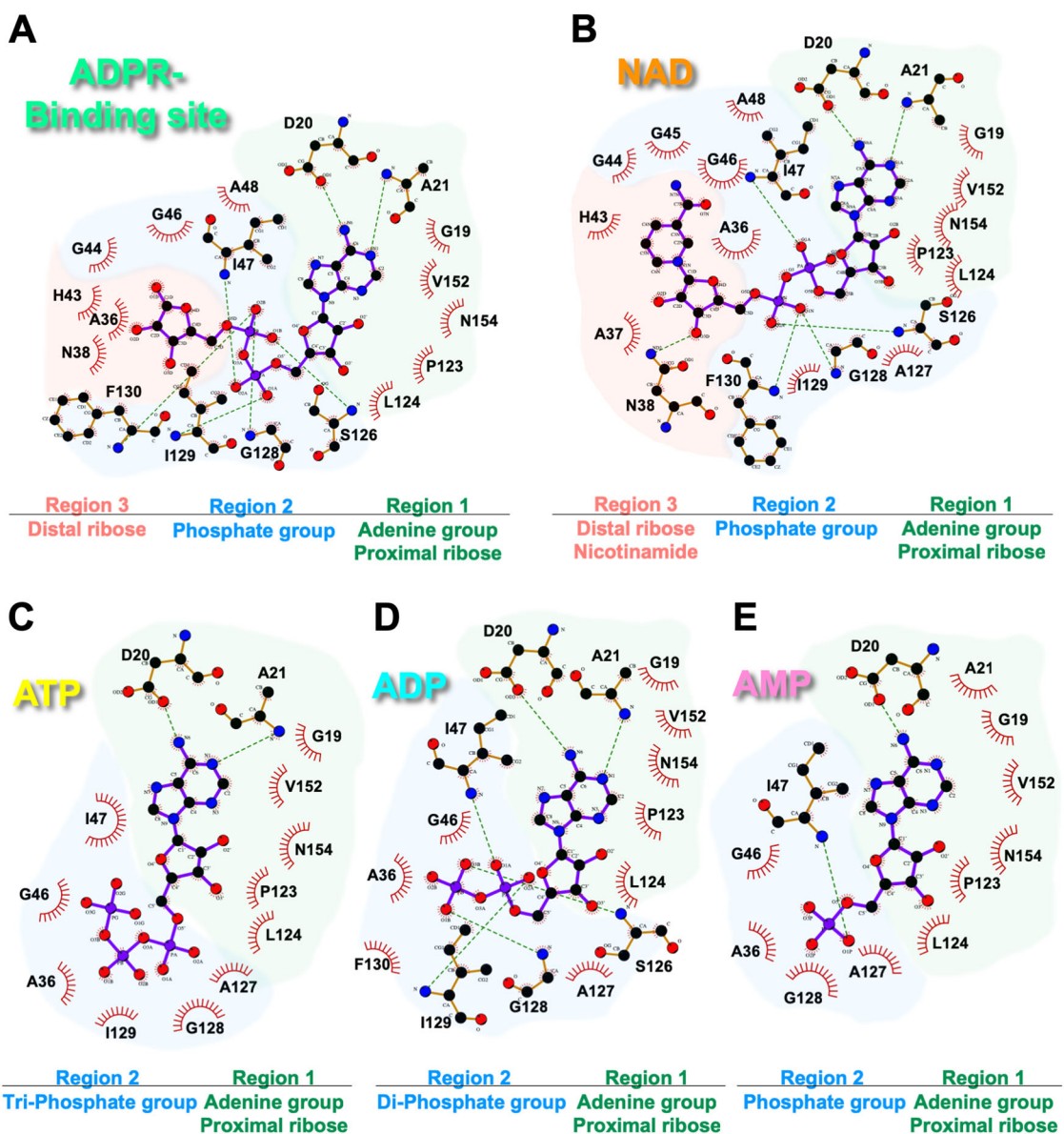

**Fig. 3 Ligplot diagrams for NAD metabolites coordinated with the MERS-CoV macro domain.** Components of **A** ADPR-binding, **B** NAD-binding, **C** ATP-binding, **D** ADP-binding, and **E** AMP-binding sites are shown. According to the chemical structure of each ligand, the binding sites are classed in several regions: region 1 in green, region 2 in marine, and region3 in salmon. Covalent bonds of ligands are in purple, covalent bonds of amino acids are in brown, and hydrogen bonds are in green dashed lines. Protein residues in hydrophobic contacts with NAD metabolites are represented by eyelash symbols.

MERS-CoV macro domain in apo form[56] was used as an initial reference for the assignment after NAD metabolite titration. Because a slow-exchange effect indicating high-affinity binding was immediately detected upon ADPR titration (Supplementary Fig. S6A), the triple resonance backbone assignment of the ADPR-complex MERS-CoV macro domain was completed to further confirm this. Otherwise, perturbations of ATP, ADP, and AMP upon the MERS-CoV macro domain were in fast-exchange and could be traced by peak migration. Surprisingly, only a few residues presented CSPs upon NAD titration toward the MERS-CoV macro domain (Supplementary Fig. S6).

The chemical shift changes were then mapped onto the surface of corresponding structures. Residues of the MERS-CoV macro domain near the adenine group exhibited CSPs in all the complex structures (Supplementary Fig. S7). Except for NAD, the residues surrounding the binding site presenting significant CSPs were consistent with the observation in crystal structures (Fig. 4A). ITC measurements confirmed a 1:1 stoichiometry of protein and

ligand binding. The NMR titration data show a slow-exchange process between apo and the ADPR-bound macro domain (Supplementary Fig. S6A), with a relatively strong dissociation constant Kd (3.67 ± 2.03 μM) obtained by ITC measurement. At 298 K, the thermodynamic profile of ADPR binding, with favorable enthalpy and unfavorable entropy, suggested the formation of a complex via hydrogen bonds mainly and a loss of conformational freedom associated with binding[46]. However, the bindings of other NAD metabolites showed less favorable enthalpy and opposite entropy contribution, resulting in a 30-fold to ≥100-fold decrease in the binding constant (Table 3: ADPR, ~3.67 μM; ATP, ~289.30 μM; ADP, ~70.29 μM; AMP, ~191.23 μM; NAD, >1000 μM). The favorable entropy may be contributed by not filling the binding pocket (ADP and AMP) and less restrictions on the rotational and translational degrees of freedom at the binding interface (ATP and NAD) (Supplementary Fig. S8). Notably, titration of the MERS-CoV macro domain had only a minor influence on the ${}^{1}$H, ${}^{15}$N resonances of the protein, which

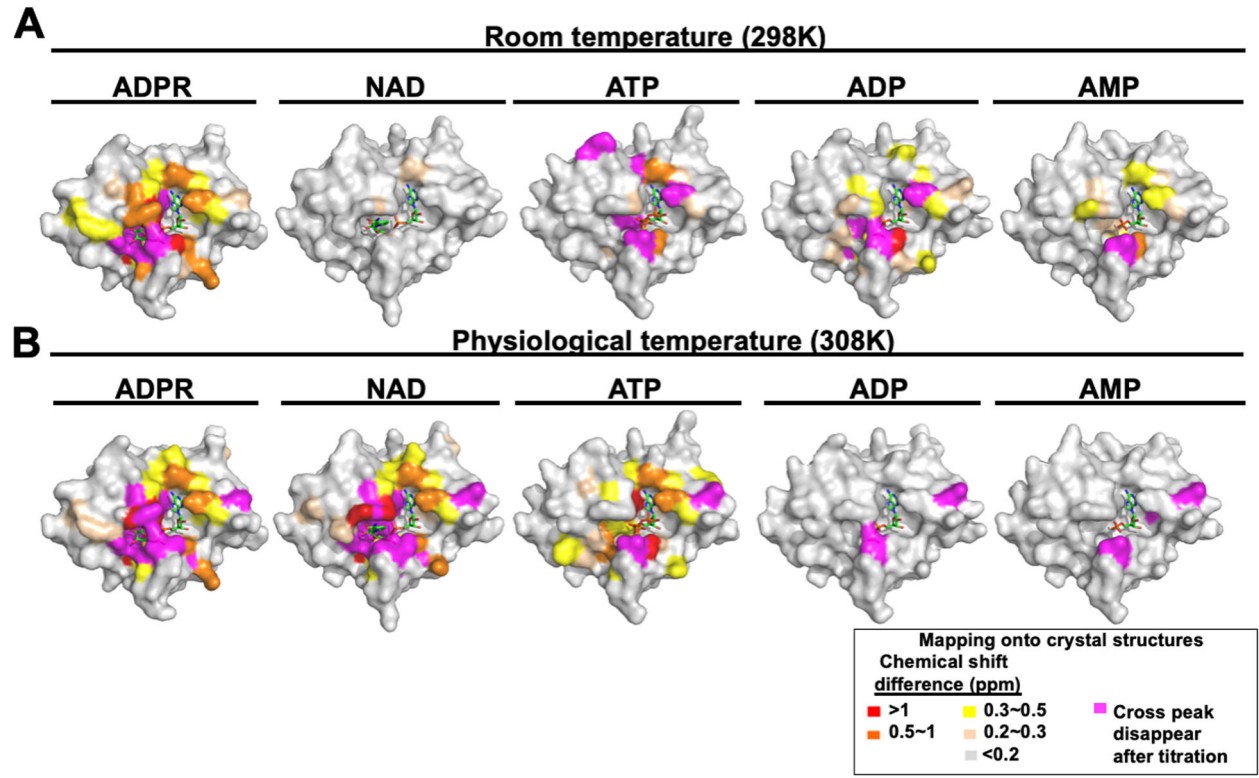

**Fig. 4 Surface mapping of the MERS-CoV macro domain with significant chemical shift changes observed in NMR perturbation at 298 K (A) and 308 K (B).** Surface colors of the MERS-CoV macro domain in complex with various NAD metabolites according to chemical shift difference upon ligand titration.

**Table 3 ITC measurements of NAD metabolites bound to MERS-CoV macro domain.**

| Ligand bound at 298 K | n | Kd (μM) | ΔH (KJ/mol) | −TΔS (KJ/mol) | ΔG (KJ/mol) |
|---|---|---|---|---|---|
| ADPR | 1.25 ± 0.08 | 3.67 ± 2.03 | −64.21 ± 4.00 | 32.85 ± 2.96 | −31.37 ± 1.69 |
| NAD | 0.938 ± 0.13 | >1000 | 1.18 ± 1.09 | −15.9 ± 1.09 | −17.12 ± 0.00 |
| ATP | 1.28 ± 0.23 | 289.30 ± 32.28 | −0.431 ± 0.26 | −19.77 ± 0.48 | −20.20 ± 0.28 |
| ADP | 1.08 ± 0.06 | 70.29 ± 29.00 | −16.11 ± 7.54 | −7.77 ± 6.32 | −23.88 ± 1.23 |
| AMP | 1.05 ± 0.38 | 191.23 ± 67.29 | −11.19 ± 3.02 | −10.12 ± 2.39 | −21.31 ± 0.81 |
| Ligand bound at 308 K | n | Kd (μM) | ΔH (KJ/mol) | −TΔS (KJ/mol) | ΔG (KJ/mol) |
| ADPR | 1.20 ± 0.19 | 26.92 ± 13.47 | −18.05 ± 18.45 | −9.30 ± 17.61 | −27.36 ± 1.33 |
| NAD (WT) | 1.04 ± 0.12 | 54.34 ± 2.81 | −2.73 ± 3.66 | −22.59 ± 3.77 | −25.32 ± 0.13 |
| NAD (G45V) | 1.31 ± 0.10 | 154.50 ± 42.19 | −76.05 ± 41.49 | 53.35 ± 41.50 | −22.70 ± 0.71 |
| NAD (I129F) | ND | | | | |
| ATP | ND | | | | |
| ADP | ND | | | | |
| AMP | ND | | | | |

*ND* not determinable.

indicates that NAD exhibits the weakest interaction among the five NAD metabolites at 298 K. The ITC results of the five NAD metabolites were consistent with the observation by NMR (Supplementary Fig. S7).

**Binding behavior of NAD metabolites at physiological temperature.** Because we observed the unexpected phenomenon of little binding for NAD in the isothermal experiments at 298 K (both NMR and ITC methods). We then checked the effect of continuous temperature change on protein by CD. The sigmoidal thermal denaturation profiles of the macro domain with various NAD metabolites reflected a cooperative protein unfolding pathway. In the presence of AMP, ADP, and ATP, the MERS-CoV macro domain showed a slight loss of CD signal before

reaching the Tm value, which suggests a compact losing but still folding structure (Fig. 1C), whereas the binding of ADPR seemed to stabilize the MERS-CoV macro domain through a conformational change because of a flat CD signal line before the Tm value (Fig. 1C, green). The MERS-CoV macro domain showed a significant Tm increase in the presence of NAD (Fig. 1B, C); however, the curve of the macro domain with NAD presented a gentle slope but similar Tm values as the protein with ADPR (Fig. 1C). This phenomenon indicated that the increase in temperature might enhance the binding affinity of NAD due to a conformational change of the MERS-CoV macro domain (Fig. 4).

To test the hypothesis, NMR CSP and ITC experiments performed at 298 K were re-executed at physiological human body temperature (308 K). The results showed almost no chemical shift perturbation of the macro domain with the

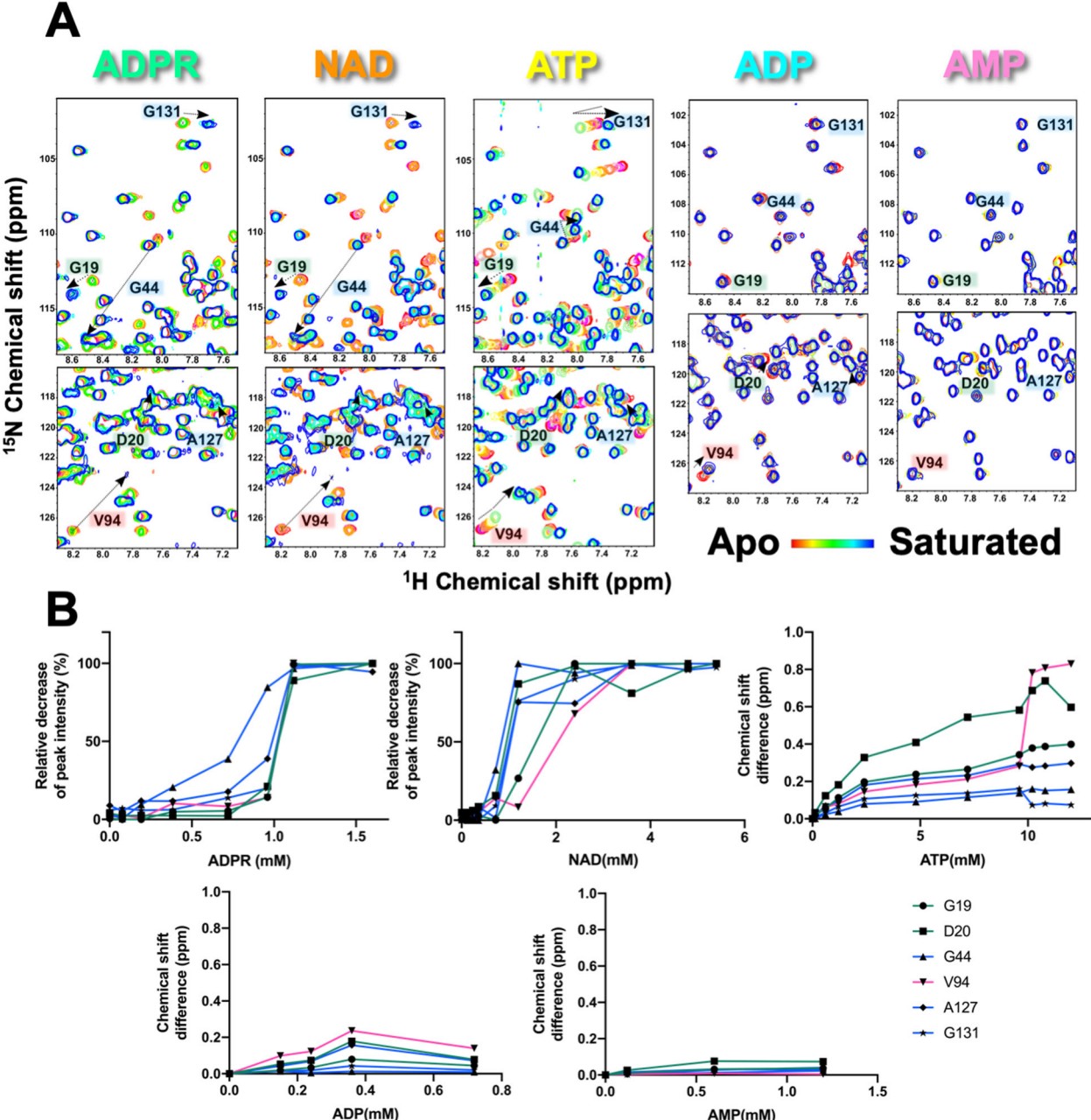

**Fig. 5 Chemical shift perturbations of residues in three regions of ligand binding site at 308 K. A** Overlays of the ¹H-¹⁵N HSQC spectra of residues in three regions of the ligand-binding site of the MERS-CoV macro domain alone and a series of NAD metabolites titrations. **B** The decrease in intensity of cross-peaks and the chemical shift difference of residues selected are plotted against the concentration of titrated NAD metabolites.

addition of AMP or ADP, and a fast-exchange on the NMR timescale of the protein with ATP titration (Supplementary Fig. S6B). The largest CSPs occurred, while undergoing chemical exchange between the free and ADPR-bound MERS-CoV macro domains, in the slow exchange regime at 308 K. Strikingly, the macro domain titrated with NAD showed a similar CSP pattern as ADPR in a slow-exchange NMR timescale at 308 K (Supplementary Fig. S7B), which indicates that the binding behavior of NAD is as strong as ADPR. In other words, the MERS-CoV macro domain is bound to both ADPR and NAD well at physiological temperature. ITC analyses also provided consistent evidence for the binding ability of these metabolites toward the MERS-CoV macro domain at physiological temperature (Supplementary Fig. S8). For binding with the other three

nucleotides (ATP, ADP, and AMP), the titration curve from ITC data could not be fitted by a binding isotherm, and no Kd value was obtained. The Kd values of the MERS-CoV macro domain at 308 K could be only determined upon ADPR and NAD titrations as 26.92 ± 13.47 μM and 54.34 ± 2.81 μM, respectively (Supplementary Fig. S8B). Moreover, this fact emphasizes the key role of distal ribose for recognizing and binding both ADPR and NAD.

At 308 K, the MERS-CoV macro domain showed significant chemical shift changes and attenuated peaks of residues at the ligand-binding cleft on titrating with ADPR or NAD (Fig. 5). The cross-peaks of some residues for three regions around the binding site were then further investigated (region 1: G19 and D20; region 2: G44, A127, and G131; and region 3: V94) (Supplementary Fig. S5). G44 was the most sensitive residue toward ADPR

titration (Fig. 5A). Sigmoid plots were fitted from the function of the relative decrease in peak intensity and the ADPR concentration, thus indicating an induced-fit binding mode of the macro domain for ADPR. On ITC measurement at 308 K, the thermodynamic profiles of the MERS-CoV macro domain upon ADPR titration showed a slight decrease in affinity (~26.92 μM) with lower favorable enthalpy ($-18.05 \pm 18.45$ KJ/mol) and favorable entropy ($-9.30 \pm 17.61$ J/mol K) contribution, comparing with the ADPR titration at 298 K. The ADPR titration results might reflect a dynamic system at 308 K. Intuitively, much less hydrogen bonding network existed at higher temperatures, although the ADPR-complexed protein persisted in the cooperatively folded state. Unexpectedly, the thermodynamic profiles of NAD showed a similar disassociation constant (~54.34 μM) but less favorable enthalpy ($-2.73 \pm 3.66$ KJ/mol) and more favorable entropy ($-22.59 \pm 3.77$ J/mol K) contribution as compared with ADPR (Table 3). Peaks of these residues were consistently perturbed during the addition of ATP at 308 K. Of note, the direction of chemical shift changes of G44 and G131, located in region 2, changed during the ATP titration (Fig. 5A), thus indicating a two-step binding for the accommodation of the terminal γ-phosphate group.

The surface mapping of the residues with significant CSPs at physiological temperature are represented (Fig. 4B). Consistent with ITC results, CSPs of NAD at 308 K had almost identical patterns with that of ADPR. During the titration of ADPR at 308 K, the residues with the most chemical shift change (>1 ppm, colored in red) and those with full attenuation (colored in magenta, Fig. 4B) located on distinguished loops 1 and 2, which contained highly conserved residues (Supplementary Fig. S1) with strong mobility. The fluctuation of these two loops at physiological temperature might yield room for a binding pocket to allow ADPR and especially NAD binding.

To further confirm our suspicion, we introduced two point-mutations into MERS-CoV macro domain, G45V, and I129F, which respectively are located at distinguished loop 1 and 2 (Supplementary Fig. S9A). The Tm differences of WT or mutant proteins Tm were monitored with the presence of various amounts of NAD by DSF thermal shift assay (Supplementary Fig. S9B). Obviously, the NAD-added G45V protein samples generated fewer Tm differences rather than the WT proteins. Moreover, there were no Tm changes in I129F groups. The NAD binding abilities of two mutants at human body temperature (308 K) were measured by ITC (Supplementary Fig. S9C–E). MERS-CoV macro domain G45V bound to NAD weakly at 308 K with the Kd around 154.5 μM. The G45V NAD-binding presented a quite different thermodynamic profile comparing with WT. The NAD binding at 308 K under the influence of G45 mutation involves more conformational changes, as indicated by the unfavorable entropy. However, the favorable enthalpy changes compensate for the unfavorable conformational changes and allow spontaneous interaction with similar $\Delta G$, as the WT. On the other hand, there was no significant heat difference detected by ITC during the NAD titrating to I129F protein.

**De-MARylation of MERS-CoV macro domain at different temperatures**. The de-mono-ADP-ribosylation (de-MAR) activity of the chikungunya virus, Hepatitis E virus, and SARS-CoV-1 has been estimated before[25,37,38]. Whether temperature influencing the MERS-CoV macro domain enzyme function, the de-MAR activity was examined at room temperature (298 K) and human body temperature (308 K). Various amounts of MERS-CoV macro domain (from 2.5 μM to 10 μM) were incubated with autoMARylated human ARTD10 catalytic domain (HsARTD10-CatD) at different temperature. The result showed that the mono-ADP-ribosylation (MAR) signals were decreased with the increase of the macro domain (Fig. 6A, B). And there was a more significant decrease of the MAR signals at 308 K rather than 298 K. Whether NAD or ADPR inhibited the deMAR function via competitive mode, the deMAR reactions were incubated with NAD or ADPR with various concentration (0.5–2.0 mM) (Fig. 6C, D). While the NAD was presenting, the MAR signals reduced more, and the signal decreases were more dramatic at 308 K than tested groups at 298 K. The MAR signal reducing levels between 308 and 298 K correlated to the NAD binding affinity at the condition of two different temperature (298 K, Kd >1000 μM; 308 K, Kd ~54.34 μM; Table 3). Furthermore, with the presence of ADPR, the MAR signals reduced fewer, suggesting that the ADPR as an in vitro deMAR competitors. Moreover, the ADPR competition ability was more robust at 298 K, which correlated to the ADPR biding ability at different temperature (298 K, Kd ~3.67 μM; 308 K, Kd ~26.92 μM; Table 3).

**Loops surrounding the ligand-binding site represent a tunable pocket**. The ADPR ITC titration results between 298 and 308 K, conformational changes with less contribution of hydrogen bonding (Supplementary Fig. S8C), might reflect a dynamic system at the higher temperature. Analysis of the B-factor of crystallographic data provides another way to describe structural flexibility. The data for the MERS-CoV macro domain in complex with a series of NAD metabolites indicated a significant increase in the mobility of distinguished loops 1 and 2 (Fig. 7A, labeled as * and +) with the exception of ADPR. Due to the orientation of the nicotinamide, the NAD-complexed MERS-CoV macro domain presented a structure with high mobile loops for distinguished loops 1 and 2 (Fig. 7B, labeled as * and +).

NMR relaxation parameters of the MERS-CoV macro domain in complex with ADPR or NAD were further measured at the magnetic field 600 MHz at 298 and 308 K (Fig. 7C, the distinguished loop 1 and 2 labeled by * and +, also colored in yellow). Distinguished loop 1 in the apo form showed a significant increase in longitudinal relaxation ($R_1$) at 308 K. Secondary structure near distinguished loop 1 (*), β3, and α2, presented high transverse relaxation ($R_2$) values. In addition, the signal decrease of NOE on the residues of distinguished loops 1 and 2 in the apo form became more pronounced, and even fell into a negative level at higher temperature. Such a result indicated these regions were undergoing dynamic motions on the nanosecond to picosecond timescale and suitable for larger ligand entry. The NMR relaxation experiments indicated that the MERS-CoV macro domain became rigid upon ADPR binding at both temperatures. Moreover, $R_1$, $R_2$ values of the distinguished loops 1 and 2 of macro domain in complex with ADPR exhibited much lower values than that of the apo form at 308 K, expect $R_2$ of G44. Remarkably, G44 of the ADPR-bound MERS-CoV macro domain presented an extremely high $J(0)$ value (Supplementary Fig. S10). The high $R_2$ and $J(0)$ of G44 in the ADPR-complexed form at 308 K might reflect the deMAR enzyme activity, since there was a suspicion that the coordination of triple-glycines at the relative position of distinguished loop 1 of VEEV regulating the deMAR ability[25].

Furthermore, we observed a striking scattering pattern of relaxation parameters for the macro domain upon NAD binding at 308 K. In addition, $R_1$ values of the NAD complex showed larger variations near distinguished loops 2 (V125, S126, and G131) and the region1 (N154 and D157) compared with that of the ADPR complex and apo form. $R_2$ values of the NAD complex showed many large variations. However, NOE values for the NAD complex showed lower variations than the apo form at distinguished loops 1 and 2. In general, the ligand-binding decreased the flexibility of protein; however, the binding of NAD

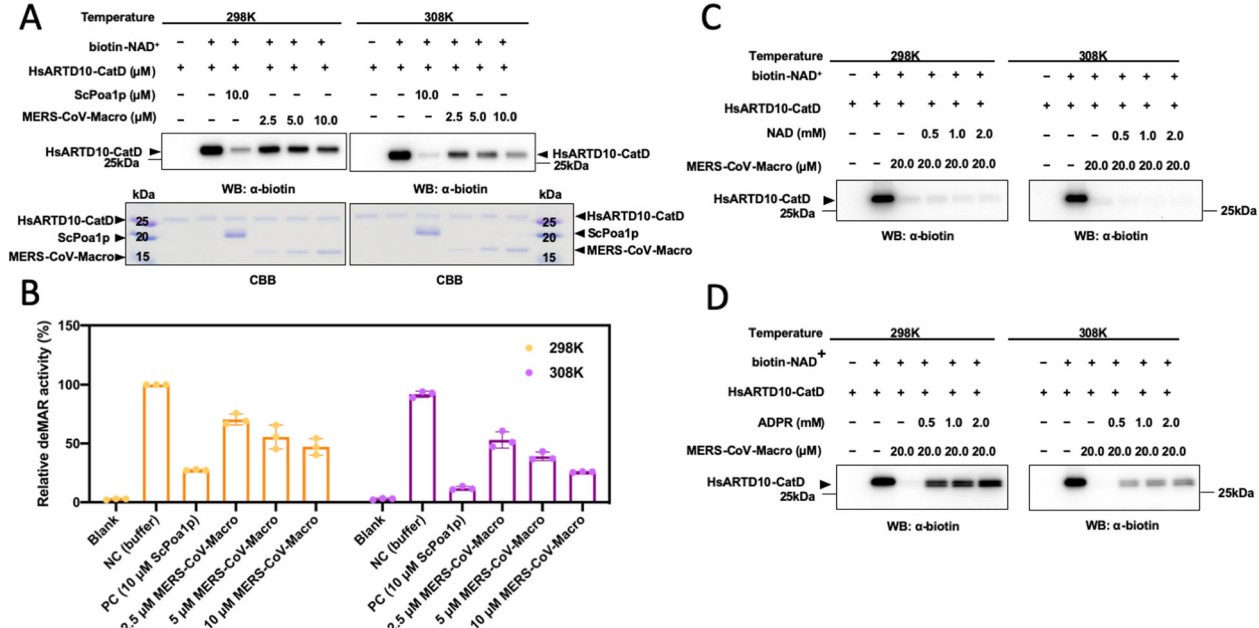

**Fig. 6 De-MARylation of the MERS-CoV macro domain at different temperatures. A** The deMAR enzyme activity of the MERS-CoV macro domain. Auto-mono-ADP-ribosylated human ARTD 10 catalytic domain (HsARTD10-CatD) was arranged using biotin-NAD$^+$ as a substrate. Budding yeast macro domain Poa1p (ScPoa1p) is a positive control. Varying amounts of the MERS-CoV macro domain (from 2.5 to 10.0 μM) mixing with biotin labeled MAR-HsARTD10-CatD at 298 and 308 K were resolved by 15% acrylamide gel. The level of mono-ADP-ribosylation was detected by western blot assay using antibody anti-biotin. **B** The relative intensity of the bands corresponding to MAR-remained after the reactions of the MERS-CoV macro domain at different temperatures. Independent experiments $n = 3$. The deMAR enzyme activity of the MERS-CoV macro domain with the presence of **C** NAD and **D** ADPR, respectively. Auto-mono-ADP-ribosylated human ARTD 10 catalytic domain (HsARTD10-CatD) was arranged using biotin-NAD$^+$ as a substrate. Varying amounts of **C** NAD or **D** ADPR mixing with 20.0 μM MERS-CoV macro domain and biotin-labeled MAR-HsARTD10-CatD at 298 K and 308 K were resolved by 15% acrylamide gel. The level of mono-ADP-ribosylation was detected by western blot assay using antibody antibiotin.

leaded high flexibility, especially near distinguished loop 2 at 308 K. Indeed, this relaxation quality of NAD-bound form reflects what correlation between the function of the MERS-CoV macro domain still needs more study.

By using molecular dynamics (MD) simulation at 298 and 308 K, we calculated the root mean square fluctuation (RMSF) for each residue in the macro domain over the 100-ns MD trajectory for the overall flexibility of the systems (Supplementary Fig. S11A). At 308 K, the macro domain showed a significant fluctuation with distinguished loop 1 (Supplementary Fig. S11A, blue box), and I129 showed a pop-up signal (blue arrow) when the temperature was enhanced. Because of the conformational change of the distinguished loops as well as the flipping of K42, H43, I129, and F130 away from the binding site at 308 K, the thermal-induced flexibility of distinguished loops 1 and 2 yielded sufficient space for entry of NAD (Supplementary Fig. S11B, C). For detecting the exposure of protein hydrophobic core, we used the fluorescence probe 8-anilinonaphthalene-1-sulfonic acid (ANS). ANS fluorescence spectra with signal enhancement at 308 K indicated that the MERS-CoV macro domain exhibited a more open conformation at 308 K than 298 K, which is consistent with the results of MD simulation (Supplementary Fig. S11D). All the data support the first description of the strong binding of NAD toward the macro domain at physiological temperature, thereby indicating that the interaction between the macro domain and NAD plays a role in biological systems.

## Discussion

In the present study, we described the crystal structures of the MERS-CoV macro domain in complex with various NAD metabolites, followed by biophysical and NMR spectroscopic investigation for comparing the interaction of the protein with these metabolites. The structures presenting here provided critical insights into the preferences of the binding pocket and helped us to understand molecular recognition in detail. Our data suggest that the MERS-CoV macro domain possesses a tunable pocket with high mobility for ligand entry.

According to the complex structures of the macro domain with adenosine–analogous ligands, the mechanistic binding steps for ADPR may be proposed. The overlapped structures with various ligands revealed that open loops become closed, because of the induced-fitness of contacting with adenosine and pyrophosphate groups. Residues at region 1 could recognize adenine groups of all ligands with the carboxylic group of D20, which was reported as a critical residue for ADPR binding on the conserved position of other macro domains (Supplementary Fig. S1)[35,46]. In addition, residues at region 2 and 3 formed hydrogen bonds with the distal ribose of ADPR, which configured a stable T-shaped-like stacking with F130. Indeed, the binding affinity evaluated by both ITC and NMR titration assay in 298 K also showed a significant increase by ADPR with an additional distal ribose as compared with ADP. The fast and slow exchange effects were observed in ADP-perturbed and ADPR-perturbed experiments, respectively. By contrast, the terminal γ-phosphate group in ATP could not compromise the binding pocket for distal ribose and was led to point out the binding site (Fig. 3C and Supplementary Fig. S3C). On the whole, the triple glycine residues at distinguished loop 1 and Ser-Ala-Gly at distinguished loop 2 presented a tunable binding pocket with appropriate conformational flexibility for different ligand interactions. ITC experiments and NMR titrations revealed that the MERS-CoV macro domain could interact with ADPR, ADP, AMP, and ATP, but the affinity to NAD was extremely weak at 298 K.

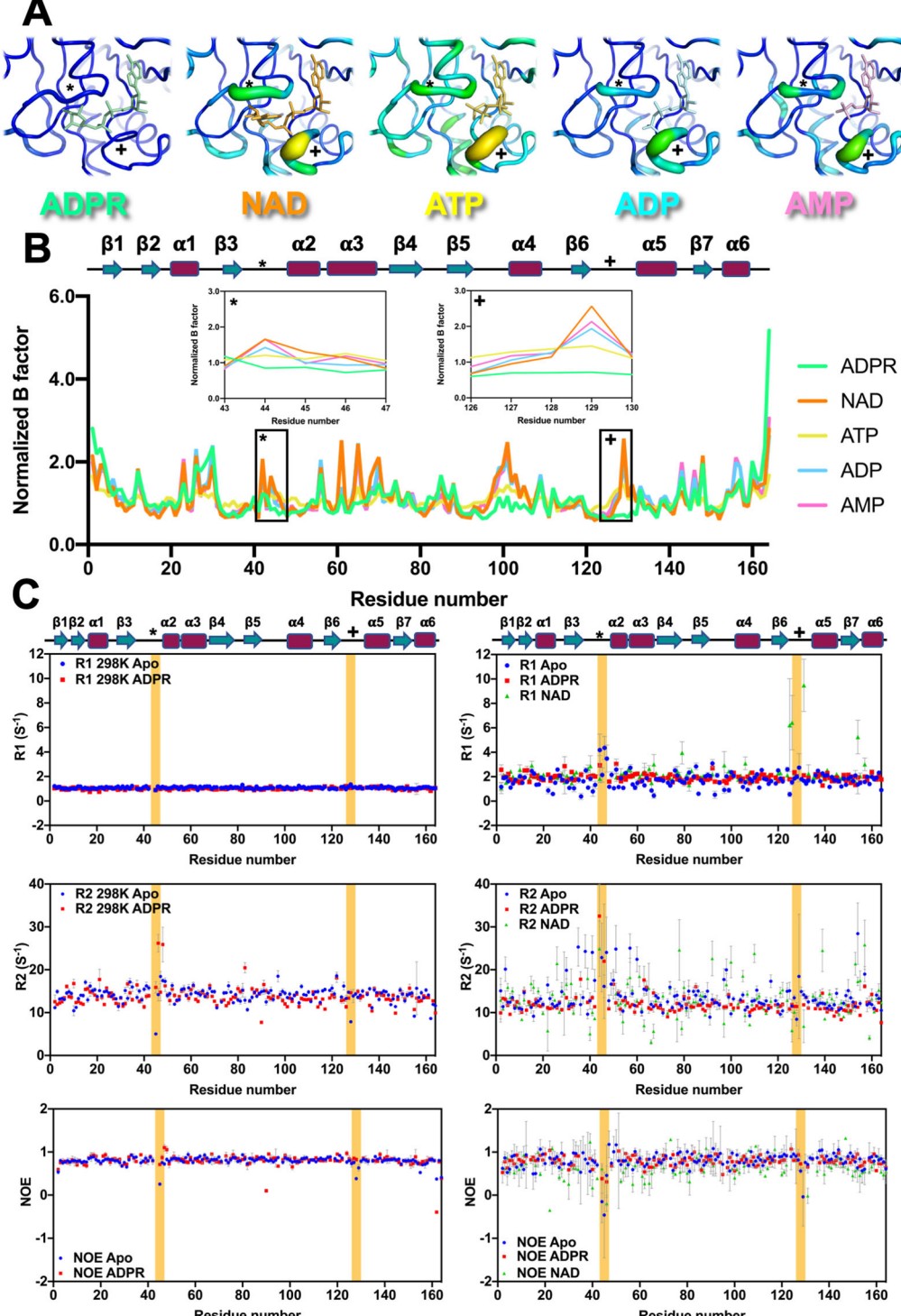

**Fig. 7 A tunable ligand-binding site of the MERS-CoV macro domain. A** Ligand binding site of MERS-CoV macro domain complexed with different NAD metabolites colored according to the B-factor putty script distributed with PyMOL. Distinguished loops are labeled by symbols, stars (*), and crosses (+). **B** Comparison of normalized B factor among various NAD metabolite-bound forms of the MERS-CoV macro domain. The flexibility difference at the distinguished loops of the ligand-binding site (*, +) is shown by insert plots of normalized B factor vs. residue number. **C** NMR relaxation parameters, $R_1$, $R_2$, NOE, of the MERS-CoV macro domain apo form, ADPR-complex, and NAD-complex form labeled by blue circles, red squares, and green triangles, respectively. The left and right panel show relaxation parameters determined at 298 K and 308 K, respectively. Parts of distinguished loops (*, +) are colored in yellow.

Binding sites in the ADP-bound, AMP-bound, and ATP-bound structures of the MERS-CoV macro domain contained fewer hydrogen bonds connecting with ligands than those in ADPR-bound structures, which suggests that the MERS-CoV macro domain possesses sequential interactions with NAD metabolites (Supplementary Fig. S4). Also, the relatively higher B-factor of loops around the binding site than other regions in these complex structures suggests that different ligand-binding properties may result from structural flexibility within the MERS-CoV macro domain (Fig. 7A, B). Recently, the macro domain of *Mayaro* virus also was found to bind with these adenosine-like nucleotides but not NAD by ITC and NMR approaches at 293 K[47].

The most interesting observation in our study is the behavior of the MERS-CoV macro domain bound to NAD. NAD has an extra nicotinamide group at its distal ribose as compared with ADPR and was unlikely to bind to the macro domain at 298 K by both ITC and NMR measurements (Table 3 and Fig. 4). However, in the crystal structure, the macro domain can accommodate the NAD molecule, with the loop near the nicotinamide group of NAD presenting little conformational difference, which suggests still a possible interaction of the MERS-CoV macro domain with NAD, this protein crystal was soaked in NAD 5 h (Fig. 3). Tm measurement of the MERS-CoV macro domain based on a continuous heating process by CD showed a temperature-dependent signal decrease in the region before Tm value. The pattern of CD curves indicates the structure of the MERS-CoV macro domain can be slightly tuned for ligand binding with increasing temperature. Therefore, we measured the binding affinities of the macro domain with NAD metabolites at physiological 308 K. The Kd value decreased for NAD binding to the macro domain at a higher temperature. Intriguingly, the affinity of NAD toward the macro domain significantly increased at physiological temperature.

We also confirmed that the tunable distinguished loops 1 and 2 are essential for NAD binding at 308 K by mutagenesis, G45V, and I129F (Supplementary Fig. S9). Remarkably, residues G45 and I129 are highly conserved in viral macro domains (Supplementary Fig. S1). NMR relaxation experiments (Fig. 7 and Supplementary Fig. S10) and MD simulation (Supplementary Fig. S11) provided evidence showing adequate flexibility of the binding pocket to accommodate ADPR-derived molecules. Finally, we identified the best-matched ligand-induced fitness at distinguished loops 1 and 2 of the MERS-CoV macro domain (the closed state of binding pocket introduced by the rotation of I129). Indeed, besides the flexibility of protein, we should consider the flexibility of ligand NAD. In the NAD-complexed crystal structure, the distal ribose of NAD was anchored, leading the nicotinamide pointed out of the binding pocket (Fig. 3B and Supplementary Fig. S3B). Although the NAD-soaking pretreatment before collecting x-ray diffraction data was at the same temperature (room temperature at NSRRC), the soaking time was extended to 5 h. So that we successfully captured the NAD-bound snapshot in the x-ray crystal structure. Our data indicate that the MERS-CoV macro domain adopts a two-step binding strategy to interact with ligands, even NAD, at physiological human body temperature: recognizing the ligand by using a tunable pocket and grabbing the ligand by an induced-fit mechanism. These ligand-induced motions give the protein a tunable binding pocket to fit ligands, a single-free ADPR, and ligand with an extra chemical structure than ADPR, for example, NAD or even PAR and ADP-ribosylated residues.

Primate macro domains containing ARTDs are evolving very rapidly with positive selection in their macro domains, for an "arms race" between these ARTDs and viral macro domains[57]. However, the function(s) of the viral macro domain remains a mystery. A model for this arms race has been proposed: ARTDs might trigger antiviral effects by recognizing the ADP-ribosylated viral or host protein via the primate macro domain. However, the effects could be blocked by removing ADPR or competition by the viral macro domain[57]. A susceptive ADPR modification cleavage model of the macro domain from VEEV was previously reported[25]. The strong dynamic pattern found in NAD binding the MERS-CoV macro domain (Fig. 7C) might be associated with the possible enzyme activity but needs more evidence. Our study found that the MERS-CoV macro domain binding to ADPR and NAD (as the important co-factor of ARTDs and sirtuins) by its tunable binding site at physiological temperature. The more NAD presenting, the more the MAR signals were reduced, and the signal decreases were more dramatic at 308 K than tested groups at 298 K. Furthermore, the more ADPR, the MAR signals reduced fewer. (Fig. 6D). The results indicated the ADPR is a competitor of the MERS-CoV macro domain as a deMAR enzyme. However, we could not conclude the NAD as an inducer of the MERS-CoV macro domain during the deMAR reaction, because the tested reaction system including HsARTD10-CatD, biotin-NAD, MERS-CoV macro domain, and NAD was stopped at the same time. To further reveal the possible inducer role of NAD, one shell creates a more precise reaction control. Nevertheless, this discovery might suggest a possible biological function of the viral macro domain is competing for NAD metabolites or blocking ADP-ribosylation signaling.

Also, we found the mobile distinguished-loops 1 and 2 at region2 of the MERS-CoV macro domain critically regulated the interaction with ADPR or NAD. This finding suggests that MERS-CoV evades host immunity by its macro domain to bind ADPR, PAR, ADP-ribosylated proteins, or NAD. Sirtuins followed a sequential binding mechanism: binding to a substrate before the co-factor NAD. Also, the NAD binding affinities of many sirtuins are extremely weak in the absence of acetylated substrates[58–60]. NAD depletion during viral infections also supports this aspect[12,61]. Recently, a sirtuin activator, resveratrol[62–64], was found to inhibit the replication of MERS-CoV[65]. The MERS-CoV macro domain could hijack cellular NAD and silence the antiviral function of sirtuins; however, the resveratrol could awake sirtuins. We need more in vivo evidence about the competitions of NAD by viral macro domains to block the antiviral properties of human proteins. Crosstalk between ARTDs and sirtuins has been reported: the activation of sirtuins might result in ARTD deacetylation, which would inhibit the activity of ARTDs[66]. Optimistically, the mentioned conflict between ADP-ribosylation and viral replication is undeniable. Macro domains are highly conserved among many positive-strand RNA viruses, which threaten global health (Supplementary Fig. S1). During the revision of this paper, the COVID-19 pandemic caused by SARS-CoV 2 appeared. Many researchers are working hard about the COVID-19, to increase the success possibility of human, we think it worthwhile to develop a pan-viral drug base for the inhibition of viral macro domains.

## Conclusion

We demonstrate an interaction between NAD metabolites with the MERS-CoV macro domain at room temperature and physiological temperature. This research is the first investigation of the ligand-binding ability of a macro domain under human body temperature. Our data indicate a ligand-binding mechanism with division and cooperation of residues near the cleft of the binding site. This binding site could be classified as three regions: region 1, near the adenine group of the ligand; region 2, constituted by distinguished loops 1 and 2 with a critical role in ligand binding; and region 3, near the distal ribose of the ligand. Region 1 plays a

role of ligand recognition. Flexibility enhancement of loops at region 2 resulted in a mobile binding site that allowed a larger ADPR-like ligand, such as NAD. The open/closed form of the distinguished loops-controlled ligand-in and ligand-out. Hydrogen bond formation between region 3 and the distal ribose was the final step to complete the ligand-induced fit mechanism. Our study provides new insights into the ligand binding mechanism of the MERS-CoV macro domain and also to what extent viral macro domains interfere in the host immunity by fluctuating the balance of the pool of NAD metabolites by interaction with its ligands.

## Methods

**Protein expression and purification.** Protein expression and purification of the MERS-CoV macro domain were as described previously[46]. Briefly, *Escherichia coli* BL21 (DE3) cells were transformed with the plasmid of PET-28a (+) containing the inserted sequence. The cells were grown at 37 °C up to $OD_{600}$ 1.0 with antibiotic kanamycin 50 μg/ml. The recombinant MERS-CoV macro domain with an N-terminal His-tag was induced by the addition of 1 mM IPTG, then grown for 20 h at 16 °C. The recombinant protein was purified using a Ni-NTA column and eluted by elution buffer (25 mM phosphate, 100 mM NaCl, pH 7.0, and 300 mM imidazole). After removing His-tag by using thrombin. The protein sample was further purified by gel filtration chromatography with a Superdex75 XK 16/60 column (GE Healthcare) in 20 mM Tris-HCl buffer (pH 7.0), 100 mM NaCl.

**Crystallization and data collection.** The MERS-CoV macro domain (10 mg/mL) was mixed with various NAD metabolites in a molar ratio of 1:16. Protein crystallization trials were performed at 283 K by the sitting-drop vapor diffusion method with commercial crystallization screen kits, 96-well Intelli-plates (Art Robbins Instruments) and a Crystal Phoenix robot (Art Robbins Instruments). The crystals for data collection were grown for 1 week at 283 K with 1.6 mM sodium citrate acid, pH 7.5. Before data collection, crystals were soaked in mother liquor with cryoprotected 20% glycerol and were flash-frozen in liquid nitrogen at 100 K. The diffraction images were collected at 100 K by using a nitrogen gas stream of BL15A1, BL13B1, or BL13C1 beamlines at the National Synchrotron Radiation Research Center, Taiwan, and processed by using HKL2000 software[67].

**Structure determination and refinement.** The crystal structures of the NAD metabolite-bound MERS-CoV macro domain were solved by molecular replacement by using Phaser[68] in the PHENIX package[69], with coordinates of the macro domain/ADPR structure from the PDB entry 5DUS[46]. The initial structure was refined with iterative cycles of simulated annealing, energy minimization, and manual rebuilding by using PHENIX refinement[69] and COOT[70]. Molecular visualizations were generated with PyMOL[71]. Data collection and refinement statistics are summarized in Table 2.

**Circular dichroism (CD) spectroscopy.** CD spectra were measured with 10 μM protein samples in buffer 20 mM phosphate buffer, pH 2.5–11.0, which was placed into a 1-mm pathlength cuvette and recorded on a JASCO J-810 spectrometer. The thermal transition of protein samples with the amount of 1 mM ADPR, NAD, ATP, ADP, and AMP were monitored at 208 nm from 10 to 95 °C at a scan rate of 1 °C/min. The melting temperature (Tm) was determined with the first derivative of the CD signal.

**Differential scanning fluorimetry (DSF).** Thermal shift assay with DSF involved the use of a CFX48 Real-Time PCR Detection System (Bio-Rad, Hercules, CA, USA) and StepOne Real-Time PCR Detection System (Thermo Fisher). Twenty-five microliter of mixture containing 2.5 μL SYPRO Orange (Sigma-Aldrich), 10–200 μM protein sample, and at most 100-fold concentrations of NAD metabolites were mixed in 8-well PCR tube. For DSF titration assays, various concentrations of NAD metabolites were used. Fluorescent signals were measured from 10 to 95 °C (excitation, 450–490 nm; detection, 560–580 nm). Data evaluation, Tm determination, and data fitting for dissociation constant (Kd) calculation were performed using Prism 8.

**Isothermal titration calorimetry.** Binding of various NAD metabolites to the MERS-CoV macro domain was measured by ITC with the Nano Isothermal Titration Calorimeter (TA Instruments). Aliquots of 4 μL of 7.5 mM ATP, 12 mM ADP, or 15 mM AMP were titrated by injection into protein (0.3 mM, 0.114 mM, or 0.6 mM in 0.98 mL, respectively) in 20 mM Tris-HCl buffer, pH 7.0 with 100 mM NaCl at 25 °C (298 K) with 250 rpm stirring. For experiments at physiological temperature, aliquots of 4 μL of 1 mM ADPR, 2 mM NAD, 6 mM ATP, 8 mM ADP, or 6 mM AMP were titrated by injection into protein (0.2 mM in 0.98 mL) in 20 mM Tris-HCl buffer, pH 7.0 with 100 mM NaCl at 25 °C (298 K). Background heats from ligand to buffer titrations were subtracted. The stoichiometry of the

binding ($n$) and the dissociation constant (Kd) were derived by fitting with an independent binding model with the use of Launch NanoAnalyze v2.3.6.

*MD simulations.* The MERS-CoV macro domain crystal structure underwent MD simulations to analyze the thermal-induced conformational change of the ligand-binding pocket at 298–308 K with the use of GROMACS 5.1.3[72]. The molecular topology of the MERS-CoV macro domain was created according to the parameters from the GROMOS96 43 a1 force field. During simulations, all protein atoms were located in a cubic water box of SPC/E molecules that extended 10 Å from the protein. $Na^+$ and $Cl^-$ ions were added into the systems and reached an ionic strength of 0.15 M. Energy minimization involved the descent algorism for 50,000 steps. The energy equilibration of the systems involved NVT and NPT ensemble for 100 ps. The temperature of the systems reached the target value (298 and 308 K) and remained stable. Upon completion of the two equilibration phases, followed by 100-ns production of MD simulations with time step 2 fs at constant pressure (1 atm) and target temperatures. The PME algorithm calculated the electrostatic interactions, and all bonds were constrained by using the LINCS algorithm. A cut-off distance used for long-range interactions was 1 nm for van der Waals and 1 nm for electrostatic interactions[73]. In this method, short-range interactions are calculated at each step of the simulation, whereas interactions at a longer distance are calculated only at each update of the non-bonded pair list and kept constant up to the next update. The reference temperature for coupling was controlled via the v-rescale coupling algorithm, and a the Parrinello–Rahman algorithm maintained a pressure of 1 atm. Convergence was checked by monitoring the RMSD deviation during the final 50 ns. Snapshots were collected every 2 ps and stored for analysis of MD simulations.

*NMR spectroscopy and chemical shift perturbation.* NMR protein samples were prepared in 25 mM phosphate buffer, pH 6.5 with 150 mM NaCl, in 90% $H_2O$/10% $D_2O$. All NMR experiments were recorded at 298 or 308 K on a Bruker AVANCE 600 spectrometer equipped with 5 mm triple resonance cryoprobe and Z-gradient. The ligand titrations were collected until the saturation of chemical shift differences. The final protein: ADPR ratios at 298 and 308 K were 1:1.76 and 1:4.5, respectively. The final protein: NAD at 298 and 308 K were 1:2.3 and 1:5.6. The final ratios of protein: ATP at 298 and 308 K were 1:72 and 1:73. The final ratios protein: ADP at 298 and 308 K were 1:36 and 1:1.2. The final protein: AMP at 298 and 308 K were 1:80 and 1:2.4. Chemical shift differences between the backbone amide $^1H$ and $^{15}N$ resonances of the MERS-CoV macro domain in apo form versus bound form were calculated by the equation[74] below.

$$\Delta\delta = \sqrt{(\Delta\delta^1 H)^2 + 0.1\left[(\Delta\delta^{15}N)^2\right]}$$

Backbone assignment of ADPR complex the MERS-CoV macro domain involved using 2D $^1H$-$^{15}N$ HSQC, CBCA(CO)NH and HNCACB spectra with the software CARA[75]. The chemical shift details of the MERS-CoV macro domain saturatedly interacting with ADRP, NAD, ATP, ADP, and AMP at 298 and 308 K were deposited to biological magnetic resonance bank with BMRB codes as 50393, 50394, 50395, 50396, and 50397, respectively.

*NMR relaxation analyses.* All NMR relaxation experiments were recorded at 298 and 308 K on a Bruker AVANCE 600 MHz spectrometer. 2D $^1H$-$^{15}N$ HSQC spectra for $^{15}N$ $R_1$, $^{15}N$ $R_2$, and $^1H$-$^{15}N$ NOE measurements were acquired. $R_1$ values were measured in a series of spectra with relaxation delays of 10, 600, 200, 70, 140, 400, 500, 1000, 300, and 800 ms. $R_2$ measurements were taken with relaxation delays of 16.96, 84.8, 169.6, 0, 203.52, 50.88, 118.72, 152.64, 254.4, and 237.43 ms. For evaluating $^1H$-$^{15}N$ steady-state NOE values, two different datasets with and without an initial proton saturation were measured. The proton saturation period was 4 s. All NMR data were processed with the software Topsipin3.2 and analyzed by using Dynamicscenter2.5.1.

*ANS fluorescence experiment.* 8-Anilinonaphthalene-1-sulfonic acid (ANS) is a fluorescence probe to detect the exposure of protein hydrophobic core[76]. Fresh 10 mM ANS was prepared by dissolving in methanol and added to the protein solution to a final concentration of 40 μM. The protein solutions with a final concentration of 0.2 mM macro domain were prepared with or without 0.15 mM guanidine hydrochloride. A sample without protein and guanidine hydrochloride was a control. The excitation wavelength was 360 nm, and spectra were respectively recorded in a region of 400–600 nm by temperature 298 or 308 K.

**DeMARylation activity assay.** To perform mono-ADP-ribosylation (MAR), reaction mixture containing 10 μM hARTD10-CatD and 100 μM Biotin-$NAD^+$ in 20 mM Tris-HCl pH 8.0, 100 mM NaCl, and 0.5 mM DTT was incubated at 30 °C for 30 min. Varying concentration of MERS-CoV-MD (from 2.5 to 10 μM) was then added into the reaction followed by incubation at 25 or 35 °C for another 30 min. The reaction was terminated by adding 1% SDS and proteins were resolved in 12% acrylamide gel. The gel was transferred onto a PVDF membrane (Bio-Rad) and probed with an antibiotin polyclonal antibody (Bethyl Laboratories Inc.) (1/5000). For competition assays, the indicated concentration of NAD or ADPR (from 0.5 to 2 mM) was added into the reaction mixture as described above with 20 μM MERS-CoV-MD by incubation at 25 or 35 °C for 30 min. The reaction was

terminated by adding 1% SDS and immunoblotted with an anti-biotin polyclonal antibody.

**Statistics and reproducibility**. For DSF thermal shift assay, deMARylation activity assays, and isothermal titrations, the experiments were performed at least three times.

**Reporting summary**. Further information on research design is available in the Nature Research Reporting Summary linked to this article.

## Data availability

The Uniprot accession number of MERS-CoV macro domain protein sequence is K9N638. The atomic coordinates and structure factors for MERS-CoV macro domain in complexes with AMP, ADP, ATP, and NAD (codes 5ZU7, 5ZU9, 5ZUA, and 5ZUB, respectively) have been deposited in the Protein Data Bank (http://wwpdb.org/). The chemical shift details of the MERS-CoV macro domain saturatedly interacting with ADRP, NAD, ATP, ADP, and AMP at 298 and 308 K were deposited to biological magnetic resonance bank with BMRB codes as 50393, 50394, 50395, 50396, and 50397, respectively. The uncropped blots and gels are provided in Supplementary Fig. S12. The source data for charts in the main figures is provided in Supplementary Data 1.

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

## Acknowledgements

This work was supported by the Ministry of Science and Technology, Taiwan (MOST 108-2628-B-002-013, MOST 109-2628-B-002-037, MOST 108-2113-M-002-011, and MOST 109-2113-M-002-003) and National Taiwan University (NTU-CC-L893501 and NTU-109L7734). We thank the Technology Commons in College of Life Science and Center for Systems Biology, National Taiwan University, for instrumental support of protein crystallization. We also thank the experimental facility and the technical services provided by the "Synchrotron Radiation Protein Crystallography Facility of the National Core Facility Program for Biotechnology, Ministry of Science and Technology" and the "National Synchrotron Radiation Research Center", a national user facility supported by the Ministry of Science and Technology, Taiwan. The NMR spectra were obtained at the High-Field Nuclear Magnetic Resonance Center (HF-NMRC) and GRC NMR Core Facility in Academia Sinica, Taiwan. The authors thank Laura Smales for copyediting the manuscript.

## Author contributions

C.H.H. conceived the study. M.H.L. performed purification of the enzyme, Y.C.C. performed biochemical assays, M.H.L. and C.C.C. performed CD, DSF, and ITC assays. M.H.L., C.C.C., and C.Y.C. performed crystallization and collected X-ray data. M.H.L. and C.H.H. determined and analyzed the crystal structure. M.H.L., Y.P.H., and C.F.C. performed NMR chemical shift perturbation and collected relaxation data. M.H.L., Y.P.H., C.F.C., and C.H.H. analyzed and determined the NMR assignment. C.H.H. performed the MD simulation. M.H.L. and C.H.H. contributed to the manuscript writing. All authors reviewed the results and approved the final version of the manuscript.

## Competing interests

The authors declare no competing interests.
