## [Peer Review File · Communications Biology]

Reviewers' comments:

Reviewer #1 (Remarks to the Author):

Nicotinamide adenine dinucleotide (NAD) is a coenzyme, abundant in living cells and participates in a number of biochemical pathways, including adenosine diphosphate ribosylation, in which there are evidences that viruses are implicated in. Viral macro domains are protein modules, which may interfere in the host immunity, but the exact mechanism is still under investigation. The elucidation of how they tune their ability to recognize and selective bind to adenine nucleotides (ADP-r)), which are products of NAD metabolism, is of great importance. The interest on the function of the viral macro domains and the NAD metabolism during immune responses is increasing considerably.

The present study provides a detailed analysis of the MERS-CoV macro domain binding selectivity over NAD and NAD metabolites under various conditions, including physiological ones.

The work performed is substantial and technically even, the results are concise but there two main concerns.

The first one is that the main conclusion of the article is that NAD may bound to MERS-CoV macro domain under physiological conditions (308K) but not in lower temperature (298K). ADP-r, binds to macro at both temperatures, having different K_d. There are sufficient NMR and ITC data supporting this finding. Although could be more interesting to be investigated the biochemical aspect of this process through assays such as the well-established de-MARylation in different temperatures.

The authors stated that this due to the higher mobility of the macro domain at higher temperature, like 308K. However, the entire nsp3 polypeptide would exhibit a higher conformational flexibility at higher temperature; thus neighboring polypeptide modules may alter macro interaction properties or even hinder the macro ligand binding cleft. The intrinsically flexibility of loops close to the binding cleft has been already reported in the literature and it is not a new finding. Mobility (through 15N-relaxation) data may also offer new insight into the discussion but are not provided.

The second concern is that, according to the data presented in the manuscript, the macro crystal structure at room temperature (298K) is capable to accommodate the NAD molecule, the macro domain in solution under the same condition, despite its flexible loops cannot, and the same domain in higher temperature can accommodate the ligand. It is strange that at the same temperature, the macro domain, is capable to accommodate the NAD by soaking the macro crystals while in solution and given the polypeptide flexibility this is not observed. Also, at higher temperature, one should also consider the flexibility of the ligand and not only the flexibility of the polypeptide, which is studied at ns-ps timescale. What about macro flexibility in other time scales?

Minor comments:

1) According to the authors distinguished loops 1 & 2 are flexible, but notably there is only a minor difference for hardly one residue that exhibit different (lower) R₂ and hetero-NOE values. What about the other loops' residues?

2) Is there any other evidence for the open-closed conformation of the apo macro domain? Did the authors observed double 1H-15N-HSQC peaks for the residues that exhibit different stated in open and closed conformation?

3) In general, the ligand binding in macro domains decrease the flexibility of the side-chain and led to more compact structure. How the authors explain the higher flexibility of the complexed forms (Figure 6C, right, especially in R1 and hetero-NOE diagrams) in respect with the apo-form at higher temperature (308K).

4) There is no clear explanation at the caption of Figure 6C, which panel corresponds to the 15N-relaxation data at 298K and the data at 308K. The part of the figure, where the relaxation data are illustrated should be thoroughly revised. The upper right panel has indications about the nature of the data (R1 apo; R1 ADPR,; R1 NAD), while the other panels do not.

5) Figure 6, should also illustrate the 15N-relaxation data for macro with NAD, even if the binding is weak.

- 6) There is no BMRB code provided for the macro-ADPr complex. Even if it is reported somewhere else, the authors should provide this code at experimental section.
- 7) Some references may need reformatting, e.g. #21, #22, #28, #42, #48, #55, #65,
- 8) Lines 113-114: "macro domain was altered at physiological temperature especially when binding to NAD". The authors should define more specifically the term "physiological temperature". A parenthesis having the following "(human body temperature)" it would be appropriate.
- 9) Materials and Methods/Protein expression and purification:
Line 434 change "PET-28a" to "pET-28a(+)"
- 10) Materials and Methods/NMR spectroscopy and chemical shift perturbation:
The steps of each titration or at least the final protein: ligand ratios have to be mentioned

Reviewer #2 (Remarks to the Author):

The article written by Meng-Hsuan Lin describes the functional, biophysical and structural characterization of MERS-CoV macro-domain (MD). In particular, the authors identified NAD to bind to MERS-CoV MD, with structural evidences. These results shed light on a new possible role of MERS-CoV MD in the viral replication. However, overall the writing of the article should be greatly improved. Some experiments are also suffering from lack of information or mistakes (TSA ; ITC). The possible effect of NAD binding on deribosylation should be tested. Also the authors may have to carefully reformulate the conclusions in order to stick to MERS-CoV MD instead of providing general conclusion for all the viral MD. Indeed, functional differences have already been observed among various viral MD and the authors do not provide evidences that the results they observed on MERS-CoV MD can be translated to other MD, especially among CoV, or in Togaviridae.

Major comments :

1) Full manuscript: Please check the tense and make sure about its homogeneity (especially in the introduction), conjugation, and singular/plural.

Introduction : the introduction should be reshuffled to help the reader to understand the role of PAR and NAD dependent cell response and the viral MD function in light of this cellular response. For example, it is hard to connect the sentences L82 and 84. What are the connection and goal? The functional role of MD for each virus should be described and general statements should be avoid. The structure of the MD, and the role of each structural element and residues should be properly depicted. The authors mention several the importance of a "flexible loop". What is its role? Binding ? Catalysis ? what residues in this loop is/are conserved ?

L115 : "suggests possible thermal activation for viral infection" : Do the authors suggest that there is, during the virus life cycle, significant thermal variations that would modify binding properties.

2) SARS CoV MD has deribosylation activity. Is it the same for MERS-CoV MD? If so, would NAD inhibit this function through a competitive mode? Given the current knowledge in MD functions, providing binding data without catalytic studies is a weakness. The authors should be able to answer these questions and significantly improve the impact of their study.

3) Binding experiments:

DSF/TSA : No experimental information is provided in Mat/Met. What is the concentration of the ligand? Also titration should be used if the authors wish to claim that a giv ligand is better than another one. This is especially true if the authors claim that the temperature may affect the binding.

ITC : the proposed experimental conditions are not physiological : 25°C and 100 mM NaCl, and look almost identical to the previous conditions. Thus this experiment cannot be analysed and no conclusion can be written from it.

These data are mandatory to validate the next experiments.

4) Structural information

A sequence alignment with relevant viral MDs would be useful to check the conservation of the residues the authors are presenting in the structures. The binding of NAD on MD would be a key information for the understanding of the function(s) of MD. The structure revealed binding specificities for NAD. The authors should validate the role of the key binding residues by site directed mutagenesis and assess the effect of the mutation(s) on the binding.

Some (but not exhaustive) minor comments

L52 : "Poly-ADP-ribosylation inducing viral replications". It is not clear what these words mean :

viral replication induces PARylation, or PARylation is a prerequisite for viral replication. Also replication would better as singular instead of plural.

L53 : There is now a classification for the ribosyl hydrolase, and synthetase (ARTD). The authors should refer to it and update the enzyme name.

L52-56 : The message is not clear. Maybe the authors should emphasize on the dual role (pro-antiviral role) of ribosylation ?

L73 : replace "even" by including in.

L74 : alphavirus and coronavirus are not the 2 only viral groups where MD are found. HEV and rubella should be also mentioned.

L90 : the studies mentioned evidenced the ribosylhydrolase activity of MDs, and not only suggested it ! Also the most relevant reference (Fehr, 2016) is missing in this part of the introduction. Here again, the authors should properly discriminate between corona-, alpha-viruses, HEV etc...

L93 : which mutation, what function is affected, leading to sensitivity for IFN ?

L343 : bacteria are transformed by plasmids, and not reverse.

Reviewer #3 (Remarks to the Author):

This article by Lin et al. used a set of biophysical techniques to study the binding behaviour for the MERS-CoV macro domain with NAD metabolites. The topic is of interest for the readers of Communications Biology. However, I do have a few major concerns regarding the manuscript before it can be published.

1. The details for the MD simulations were poorly described, which makes it possible to reproduce the current work. For instance, it is not clear what kind of force fields were used to describe the systems (including the different ligands). What is the ionic concentration? Some of the setting up is not proper, e.g. all the bonds were kept rigid (most likely the authors constrained all the bonds involving hydrogen); The authors mentioned twin-range but only one cut-off was given.

2. Convergence check for the MD simulations was not mentioned.

3. ITC data: The error analyses should be provided for the enthalpy and entropy analyses. Delta G for ADPR at 298 seems to be incorrect by looking at Delta H and T Delta S. What is the possible reason for such a big difference for Delta H of ADPR at 298 K vs 308K? ADPR has the most different binding compared to other ligands at 298K. What is the molecular mechanism?

We thank the Reviewers and Editor for their valuable comments. We have addressed the points in the revised manuscript. All corrections made in the revised manuscript/figure legends are marked in red. Please see the point-by-point response as follows:

Reviewers' comments:

Reviewer #1 (Remarks to the Author):

Nicotinamide adenine dinucleotide (NAD) is a coenzyme, abundant in living cells and participates in a number of biochemical pathways, including adenosine diphosphate ribosylation, in which there are evidences that viruses are implicated in. Viral macro domains are protein modules, which may interfere in the host immunity, but the exact mechanism is still under investigation. The elucidation of how they tune their ability to recognize and selective bind to adenine nucleotides (ADP-r)), which are products of NAD metabolism, is of great importance. The interest on the function of the viral macro domains and the NAD metabolism during immune responses is increasing considerably.

The present study provides a detailed analysis of the MERS-CoV macro domain binding selectivity over NAD and NAD metabolites under various conditions, including physiological ones.

The work performed is substantial and technically even, the results are concise but there two main concerns.

The first one is that the main conclusion of the article is that NAD may bound to MERS-CoV macro domain under physiological conditions (308K) but not in lower temperature (298K). ADP-r, binds to macro at both temperatures, having different Kd. There are sufficient NMR and ITC data supporting this finding. Although could be more interesting to be investigated the biochemical aspect of this process through assays such as the well-established de-MARylation in different temperatures.

A: We are very grateful to the reviewer for this point. Detailed de-MARylation investigations of the MERS-CoV macro domain at 298 and 308K were provided in Fig. 6 and the paragraph below.

“De-MARylation of MERS-CoV macro domain at different temperatures

The de-mono-ADP-ribosylation (de-MAR) activity of the chikungunya virus, Hepatitis E virus, and SARS-CoV-1 has been estimated before^{25,37,38}. Whether temperature influencing the MERS-CoV macro domain enzyme function, the de-MAR activity was examined at room temperature (298K) and human body temperature (308K). Various amounts of MERS-CoV macro domain (from 2.5 μ M to 10 μ M) were incubated with autoMARylated human ARTD10 catalytic domain (HsARTD10-CatD) at different temperature. The result showed that the mono-ADP-ribosylation (MAR) signals were decreased with the increase of the macro domain (Fig. 6A and 6B). And there was a more significant decrease of the MAR signals at 308K rather than 298K.”

The authors stated that this due to the higher mobility of the macro domain at higher temperature, like 308K. However, the entire nsp3 polypeptide would exhibit a higher conformational flexibility at higher temperature; thus neighboring polypeptide modules may alter macro interaction properties or even hinder the macro ligand binding cleft. The intrinsically flexibility of loops close to the binding cleft has been already reported in the literature and it is not a new finding. Mobility (through

15N-relaxation) data may also offer new insight into the discussion but are not provided.

A: Thanks for the comments. We described more detail about NMR relaxation data. Also, we provide further analysis of reduced spectral density mapping in Supplementary Figure S10. The data indicate distinguished loop 1 and 2 (* and +) presenting internal motions at pico-to millisecond, reflecting by the increase $J(0)$ and $J(0.87\omega_H)$ values.

The second concern is that, according to the data presented in the manuscript, the macro crystal structure at room temperature (298K) is capable to accommodate the NAD molecule, the macro domain in solution under the same condition, despite its flexible loops cannot, and the same domain in higher temperature can accommodate the ligand. It is strange that at the same temperature, the macro domain, is capable to accommodate the NAD by soaking the macro crystals while in solution and given the polypeptide flexibility this is not observed. Also, at higher temperature, one should also consider the flexibility of the ligand and not only the flexibility of the polypeptide, which is studied at ns-ps timescale. What about macro flexibility in other time scales?

A: Thanks for the comment. Via reduced spectral density mapping data, we provided in Supplementary Figure S10, indicates distinguished loop 1 and 2 (* and +) presenting internal motions at pico-to millisecond, reflecting by the increase $J(0)$ and $J(0.87\omega_H)$ values. Also, we further describe the ligand flexibility of the NAD-complex MERS-CoV macro domain in the discussion section.

Minor comments:

1) According to the authors distinguished loops 1 & 2 are flexible, but notably there is only a minor difference for hardly one residue that exhibit different (lower) R2 and hetero-NOE values. What about the other loops' residues?

A: Thanks for the comments. It seems that the unclear illustration of Fig. 7C making misunderstandings. We added more clear illustrations in the revised manuscript and described more detail about relaxation data.

2) Is there any other evidence for the open-closed conformation of the apo macro domain? Did the authors observed double 1H-15N-HSQC peaks for the residues that exhibit different stated in open and closed conformation?

A: Thanks for the comment. Indeed, we do not observe double 1H-15N-HSQC peaks for residue I129 that exhibit different stated in open and closed conformation of the apo macro domain. We suggested the binding of a proper ligand would induce the open-closed conformational changes.

3) In general, the ligand binding in macro domains decrease the flexibility of the side-chain and led to more compact structure. How the authors explain the higher flexibility of the complexed forms (Figure 6C, right, especially in R1 and hetero-NOE diagrams) in respect with the apo-form at higher temperature (308K).

A: Thanks for the comment. We discussed this point more in the revised manuscript. The higher flexibility of the ADPR-bound form may be correlated to the deMAR enzyme activity.

4) There is no clear explanation at the caption of Figure 6C, which panel corresponds to the 15N-relaxation data at 298K and the data at 308K. The part of the figure, where the relaxation data are illustrated should be thoroughly revised. The upper right panel has indications about the nature of the data (R1 apo; R1 ADPR; R1 NAD), while the other panels do not.

A: Thanks for the comment. We added the illustrations of Fig. 6C (Fig. 7C in revised manuscript).

5) Figure 6, should also illustrate the 15N-relaxation data for macro with NAD, even if the binding is weak.

A: Thanks for the comments. Considering the almost no cross-peak shifting during

the NAD titration at 298K, we do not collect the ¹⁵N-relaxation data for macro with NAD at this temperature since there was no interaction.

6) *There is no BMRB code provided for the macro-ADPr complex. Even if it is reported somewhere else, the authors should provide this code at experimental section.*

A: Thanks for the comment. We provided the BMRB code of MERS-CoV macro domain with ADPR, NAD, ATP, ADP and AMP as 50393, 50394, 50395, 59396 and 50397, respectively in NMR experiment section of materials and methods.

7) *Some references may need reformatting, e.g. #21, #22, #28, #42, #48, #55, #65,*

A: Thanks for the comment. We reformatted and fixed these references.

8) *Lines 113-114: “macro domain was altered at physiological temperature especially when binding to NAD”. The authors should define more specifically the term “physiological temperature”. A parenthesis having the following “(human body temperature)” it would be appropriate.*

A: Thanks for this suggestion. We changed the words in the revised manuscript.

9) *Materials and Methods/Protein expression and purification:*

Line 434 change “PET-28a” to “pET-28a(+)

A: Thanks for the comment. We changed the words in the revised manuscript.

10) *Materials and Methods/NMR spectroscopy and chemical shift perturbation:*

The steps of each titration or at least the final protein: ligand ratios have to be mentioned

A: Thanks for the suggestion. We provided the final protein/ligand ratios in the section of materials and methods.

Reviewer #2 (Remarks to the Author):

The article written by Meng-Hsuan Lin describes the functional, biophysical and structural characterization of MERS-CoV macro-domain (MD). In particular, the authors identified NAD to bind to MERS-CoV MD, with structural evidences. These

results shed light on a new possible role of MERS-CoV MD in the viral replication. However, overall the writing of the article should be greatly improved. Some experiments are also suffering from lack of information or mistakes (TSA ; ITC). The possible effect of NAD binding on deribosylation should be tested. Also the authors may have to carefully reformulate the conclusions in order to stick to MERS-CoV MD instead of providing general conclusion for all the viral MD. Indeed, functional differences have already been observed among various viral MD and the authors do not provide evidences that the results they observed on MERS-CoV MD can be translated to other MD, especially among CoV, or in Togaviridae.

Major comments :

1) Full manuscript: Please check the tense and make sure about its homogeneity (especially in the introduction), conjugation, and singular/plural.

Introduction : the introduction should be reshuffled to help the reader to understand the role of PAR and NAD dependent cell response and the viral MD function in light of this cellular response. For example, it is hard to connect the sentences L82 and 84. What are the connection and goal? The functional role of MD for each virus should be described and general statements should be avoid. The structure of the MD, and the role of each structural element and residues should be properly depicted. The authors mention several the importance of a “flexible loop”. What is its role?

Binding ? Catalysis ? what residues in this loop is/are conserved ?

L115 : “suggests possible thermal activation for viral infection” : Do the authors suggest that there is, during the virus life cycle, significant thermal variations that would modify binding properties.

A: We appreciated the reviewer for this kind reminding. We checked the full manuscript and modified the section of introduction as suggested. Description of structure of the macro domain and each structural element was also included in “introduction” as follow.

“In the ligand binding site of MERS-CoV macro domain, conserved D20 made a critical recognition to the nitrogen of the adenine group of ADPR. G46, I47, S126, G128, I129 and F130 of MERS-CoV macro domain formed hydrogen bonds with the diphosphate groups of ADPR. Furthermore, A37, N38, K42, G44, A48, V93 and G95 formed hydrogen bonds with the distal ribose of ADPR.”

In addition, we described more detail about the “flexible loop” to avoid misunderstanding.

About L115, as reviewer's words, we suggested that during the virus life cycle, significant thermal variations would modify binding properties.

2) SARS CoV MD has deribosylation activity. Is it the same for MERS-CoV MD? If so, would NAD inhibit this function through a competitive mode? Given the current knowledge in MD functions, providing binding data without catalytic studies is a weakness. The authors should be able to answer these questions and significantly improve the impact of their study.

A: Thanks for the comment. For answering these questions, we conducted the deribosylation assay experiments of MERS-CoV macro domain additionally. Detailed de-MARylation investigations of the MERS-CoV macro domain in different temperatures and the competition with NAD and ADPR, are provided in Fig. 6 of revised manuscript. We described these investigations in the two paragraphs below in revised manuscript.

“De-MARylation of MERS-CoV macro domain at different temperatures

The de-mono-ADP-ribosylation (de-MAR) activity of the chikungunya virus, Hepatitis E virus, and SARS-CoV-1 has been estimated before^{25,37,38}. Whether temperature influencing the MERS-CoV macro domain enzyme function, the de-MAR activity was examined at room temperature (298K) and human body temperature (308K). Various amounts of MERS-CoV macro domain (from 2.5 μ M to 10 μ M) were incubated with autoMARylated human ARTD10 catalytic domain (HsARTD10-CatD) at different temperature. The result showed that the mono-ADP-ribosylation (MAR) signals were decreased with the increase of the macro domain (Fig. 6A and 6B). And there was a more significant decrease of the MAR signals at 308K rather than 298K. Whether NAD or ADPR inhibited the deMAR function via competitive mode, the deMAR reactions were incubated with NAD or ADPR with various concentration (0.5 to 2.0 mM) (Fig. 6C and 6D). While the NAD was presenting, the MAR signals reduced more, and the signal decreases were more dramatic at 308K than tested groups at 298K. The MAR signal reducing levels between 308K and 298K correlated to the NAD binding affinity at the condition of two different temperature (298K, $K_d > 1000 \mu$ M; 308K, $K_d \sim 54.34 \mu$ M; Table 3). Furthermore, with the presence of ADPR, the MAR signals reduced fewer, suggesting that the ADPR as an in vitro deMAR competitors. Moreover, the ADPR competition ability was more robust at 298K, which correlated to the ADPR binding ability at different temperature (298K, $K_d \sim 3.67 \mu$ M; 308K, $K_d \sim 26.92 \mu$ M; Table

3).”

“The more NAD presenting, the MAR signals reduced more, and the signal decreases were more dramatic at 308K than tested groups at 298K. Furthermore, the more ADPR, the MAR signals reduced fewer. (Fig. 6C and 6D). The results indicated the ADPR is a competitor of the MERS-CoV macro domain as a deMAR enzyme. However, we could not conclude the NAD as an inducer of the MERS-CoV macro domain during the deMAR reaction, because the tested reaction system including HsARTD10-CatD, biotin-NAD, MERS-CoV macro domain, and NAD was stopped at the same time. To further reveal the possible inducer role of NAD, one shell creates a more precise reaction control. Nevertheless, this discovery might suggest a possible biological function of the viral macro domain is competing for NAD metabolites or blocking ADP-ribosylation signaling.”

3) Binding experiments:

DSF/TSA : No experimental information is provided in Mat/Met. What is the concentration of the ligand? Also titration should be used if the authors wish to claim that a given ligand is better than another one. This is especially true if the authors claim that the temperature may affect the binding.

A: We are very appreciative of the reviewer for this kind reminding. We added the information of DSF in the section of Materials and Methods. We also conducted an additional experiment as “a series of titrations monitored by DSF” in Supplementary Figure S2.

ITC : the proposed experimental conditions are not physiological : 25°C and 100 mM NaCl, and look almost identical to the previous conditions. Thus this experiment cannot be analysed and no conclusion can be written from it. These data are mandatory to validate the next experiments.

A: We are very appreciative of the reviewer for this kind reminding. We performed the ITC assay of macro domain with ADPR titration at 298K again to replace the data used in the previous publication and shown in revised Supplementary Figure S8A.

4) Structural information

A sequence alignment with relevant viral MDs would be useful to check the conservation of the residues the authors are presenting in the structures. The binding of NAD on MD would be a key information for the understanding of the function(s) of MD. The structure revealed binding specificities for NAD. The authors should validate the role of the key binding residues by site directed mutagenesis and assess the effect of the mutation(s) on the binding.

A: We are very grateful to the reviewer for this point. We provided a sequence alignment with viral macro domains in Supplementary Figure S1. As suggested, we

confirmed our points by additional mutagenesis, DSF and ITC experiments and showed in Supplementary Figure S9. Also, we described in the paragraph below.

“To further confirm our suspicion, we introduced two point-mutations into MERS-CoV macro domain, G45V, and I129F, which respectively are located at distinguished loop 1 and 2 (Supplementary Figure S9A). The T_m differences of WT or mutant proteins T_m were monitored with the presence of various amounts of NAD by DSF thermal shift assay (Supplementary Figure S9B). Obviously, the NAD-added G45V protein samples generated fewer T_m differences rather than the WT proteins. Moreover, there were no T_m changes in I129F groups. The NAD binding abilities of two mutants at human body temperature (308K) were measured by ITC (Supplementary Figure S9 C-E). MERS-CoV macro domain G45V bound to NAD weakly at 308K with the K_d around 154.5 μ M. The G45V NAD-binding presented a quite different thermodynamic profile comparing with WT. The NAD binding at 308K under the influence of G45 mutation involves more conformational changes, as indicated by the unfavorable entropy. However, the favorable enthalpy changes compensate for the unfavorable conformational changes and allow spontaneous interaction with similar ΔG , as the WT. On the other hand, there was no significant heat difference detected by ITC during the NAD titrating to I129F protein.”

Some (but not exhaustive) minor comments

L52 : “Poly-ADP-ribosylation inducing viral replications”. It is not clear what these words mean : viral replication induces PARylation, or PARylation is a prerequisite for viral replication. Also replication would better as singular instead of plural.

A: Thanks for the comment. We described this sentence more clear, and we change the “replication” as singular.

L53 : There is now a classification for the ribosyl hydrolase, and synthetase (ARTD). The authors should refer to it and update the enzyme name.

A: Thanks for this suggestion. We changed the enzyme names in the revised manuscript.

L52-56 : The message is not clear. Maybe the authors should emphasize on the dual role (pro-antiviral role) of ribosylation ?

A: We are very grateful to the reviewer for this suggestion. We described this sentence more clear.

L73 : replace “even” by including in.

A: We are very appreciative of the reviewer for this kind reminding. We changed the words in the revised manuscript.

L74 : alphavirus and coronavirus are not the 2 only viral groups where MD are found. HEV and rubella should be also mentioned.

A: We are very appreciative of the reviewer for this kind reminding. We added the HEV and rubella virus in the revised manuscript.

L90 : the studies mentioned evidenced the ribosylhydrolase activity of MDs, and not only suggested it ! Also the most relevant reference (Fehr, 2016) is missing in this part of the introduction. Here again, the authors should properly discriminate between corona-, alpha-viruses, HEV etc...

A: Thanks for this suggestion. We changed the description in the revised manuscript. And we add the reference (Fehr, 2016) at this sentence.

L93 : which mutation, what function is affected, leading to sensitivity for IFN ?

A: We are very appreciative of the reviewer for this kind reminding. We added the position of that mutation in the revised manuscript.

L343 : bacteria are transformed by plasmids, and not reverse.

A: Corrected

Reviewer #3 (Remarks to the Author):

This article by Lin et al. used a set of biophysical techniques to study the binding behaviour for the MERS-CoV macro domain with NAD metabolites. The topic is of interest for the readers of Communications Biology. However, I do have a few major concerns regarding the manuscript before it can be published.

1. The details for the MD simulations were poorly described, which makes it possible to reproduce the current work. For instance, it is not clear what kind of force fields

were used to describe the systems (including the different ligands). What is the ionic concentration? Some of the setting up is not proper, e.g. all the bonds were kept rigid (most likely the authors constrained all the bonds involving hydrogen); The authors mentioned twin-range but only one cut-off was given.

A: We are very appreciative of the reviewer for this kind reminding. We added more details about MD simulations in the revised manuscript.

2. Convergence check for the MD simulations was not mentioned.

A: We are very appreciative of the reviewer for this kind reminding. We added the convergence check in the section of materials and methods in the revised manuscript.

3. ITC data: The error analyses should be provided for the enthalpy and entropy analyses. Delta G for ADPR at 298 seems to be incorrect by looking at Delta H and T Delta S. What is the possible reason for such a big difference for Delta H of ADPR at 298 K vs 308K? ADPR has the most different binding compared to other ligands at 298K. What is the molecular mechanism?

A: We appreciated the reviewer for this kind reminding. We provided the error analyses of ITC data in the revised manuscript (Fig S8. and Table 3) and checked the delta G values in our data. Discussion of the different the difference of delta H of ADPR at 298K vs. 308K in the revised manuscript was added as the two paragraphs below.

“On ITC measurement at 308K, the thermodynamic profiles of the MERS-CoV macro domain upon ADPR titration showed a slight decrease in affinity ($\sim 26.92 \mu\text{M}$) with lower favorable enthalpy ($-18.05 \pm 18.45 \text{ KJ/mol}$) and favorable entropy ($-9.30 \pm 17.61 \text{ J/mol K}$) contribution, comparing with the ADPR titration at 298K. The ADPR titration results might reflect a dynamic system at 308K. Intuitively, much less hydrogen bonding network existed at higher temperatures, although the ADPR-complexed protein persisted in the cooperatively folded state.” “The ADPR ITC titration results between 298K and 308K, conformational changes with less contribution of hydrogen bonding (Supplementary Figure S8C), might reflect a dynamic system at the higher temperature.”

REVIEWERS' COMMENTS:

Reviewer #1 (Remarks to the Author):

The authors have substantially revised the manuscript and replied to all reviewers' comments, one by one.

New data presented now and support their original results and their conclusions.

I think that the manuscript has been substantially improved and meets now the criteria for publication in Communications Biology journal.

Reviewer #3 (Remarks to the Author):

The authors have addressed my comments.